# A host–guest semibiological photosynthesis system coupling artificial and natural enzymes for solar alcohol splitting

Junkai Cai[1], Liang Zhao [1✉], Cheng He[1], Yanan Li[1] & Chunying Duan [1✉]

Development of a versatile, sustainable and efficient photosynthesis system that integrates intricate catalytic networks and energy modules at the same location is of considerable future value to energy transformation. In the present study, we develop a coenzyme-mediated supramolecular host-guest semibiological system that combines artificial and enzymatic catalysis for photocatalytic hydrogen evolution from alcohol dehydrogenation. This approach involves modification of the microenvironment of a dithiolene-embedded metal-organic cage to trap an organic dye and NADH molecule simultaneously, serving as a hydrogenase analogue to induce effective proton reduction inside the artificial host. This abiotic photocatalytic system is further embedded into the pocket of the alcohol dehydrogenase to couple enzymatic alcohol dehydrogenation. This host-guest approach allows in situ regeneration of NAD$^+$/NADH couple to transfer protons and electrons between the two catalytic cycles, thereby paving a unique avenue for a synergic combination of abiotic and biotic synthetic sequences for photocatalytic fuel and chemical transformation.

[1] State Key Laboratory of Fine Chemicals, Zhang Dayu School of Chemistry, Dalian University of Technology, Dalian, People's Republic of China.
✉email: zhaol@dlut.edu.cn; cyduan@dlut.edu.cn

Abiotic–biotic hybrid systems that combine light-driven artificial catalysis with biosynthetic enzymes at the same location have emerged as attractive and versatile avenues for light-trap fuel and chemical transformation with high efficacy and selectivity[1–5]. Recent advances have demonstrated that coupling solar fuel synthesis with value-added dehydrogenation may enhance economic and environmental benefits sans the expense of sacrificial reagents, while avoiding the harsh conditions required for the reforming processes[6,7]. Ethanol is a promising hydrogen storage chemical that can be effectively dehydrogenated by alcohol dehydrogenase (ADH) with the assist of coenzymes[8,9]. However, due to the inherent two-electron reduction characteristic[10,11], the use of NADH (reduced nicotinamide adenine dinucleotide) to mediate artificial photoinduced proton reduction with enzymatic conversions remains a steep challenge in homogeneous system due to issues related to kinetic synergy and catalytic compatibility[12,13]. Precise matching of the kinetics of multiple electron transfer steps between abiotic and biotic components is a prerequisite to restrain the competitive reaction of NADH radical aggregation with photosensitizer radicals or itself[14,15].

Consequently, the integration of photosensitizer, coenzyme, and catalyst into one working module via the host–guest approach is promising to co-localize the essential components within the catalytic pocket of ADH and manipulate biomimetic catalysis at the molecular level[16–18]. Of the reported artificial supramolecular catalysts, metal-organic cages are superficially reminiscent of enzymes by modulating the microenvironment to accommodate and interact with substrates[19–23]. Dye-containing metal-organic cages exhibit profound effects in regulating and promoting the light-driven hydrogen evolution and related hydrogenation[24,25]. Incorporation of dye-containing metal-organic cages into the ADH catalytic pocket was expected to eliminate inherent communication barriers as well as mutual inactivation between the abiotic and biotic systems and promote the delivery of matters and energy, thereby facilitating the combination of NAD$^+$-mediated dehydrogenation and NADH-modified hydrogen evolution[12,13,26,27]. Successful realization of paradigmatic structural fitness and kinetic compatibility requires careful orchestration of organic dye encapsulated by a potential-matching redox-active metal-organic cage for inclusion into the ADH catalytic pocket in a matryoshka fashion[26,27].

Here, we report a cobalt dithiolene-embedded pillared cage capable of trapping the shape and size matching photosensitizer and the coenzyme NADH as the middle layer of matryoshka, thereby combining abiotic photocatalytic hydrogen production with biotic dehydrogenation of alcohol within the ADH catalytic pocket (Fig. 1). The coexistence of the planar dye, 2-phenyl-4-(1-naphthyl)-quinolinium (PNQ)[28], and the coenzyme NADH, within one redox-active microenvironment can enforce close proximity between these components to enhance the efficacy of photoinduced electron transfer inside metal-organic cage[29,30], while simultaneously allowing efficient photocatalytic hydrogen production to directly produce NAD$^+$ in analog to the natural hydrogenase[31]. While situated inside the ADH catalytic pocket, the coenzyme is in direct contact with two catalytic cycles in situ, which enables it to maintain a closed loop of electrons and protons, thereby allowing the formation of a versatile redox-neutral photosynthesis system to actuate a non-photoactive natural enzyme for solar chemical conversion.

## Results

### Preparation and characterization of the metal-organic cage.

The tripodal tris(benzene-o-dithiol) ligand with rich π-electron plane, H$_6$TPS ($N,N'$-(5'-(4-(2,3- dimercaptobenzamido)phenyl)-[1,1':3',1''-terphenyl]-4,4''-diyl)bis(2,3-dimercaptoben-zamide)), was synthesized through amide condensation of freshly prepared 2,3-bis(isopropylthio)benzoyl chloride and 1,3,5-tris(4-amino-phenyl) benzene, followed by removal of the protecting groups (Supplementary Fig. 1). The reaction of H$_6$TPS and Co(BF$_4$)$_2$·6H$_2$O in $N,N$-dimethylformamide (DMF) solution containing NaOH and NEt$_4$Cl yielded 51% of the blue compound Co$_3$TPS$_2$, implying the formation of cobalt dithiolene-embedded molecular cage (Fig. 1)[32]. Installing cobalt dithiolene species on the metal-organic cage was expected to endow the artificial host with negative charge while maintaining superior redox activity with a desired low overpotential[33,34], which was essential for Co$_3$TPS$_2$ to create a redox microenvironment in analog to hydrogenase and firmly bind the positively charged photosensitizer through its pocket to form host–guest species[32]. Single-crystal X-ray structural analysis of Co$_3$TPS$_2$ revealed that the redox-active cobalt dithiolene species were recast to form a molecular anionic triangular prism cage with a pseudo-$C_3$ symmetry matching well with the optimal geometrical structures by theoretical calculations (Fig. 2a, b and Supplementary Figs. 2, 33). Two deprotonated H$_6$TPS molecules bonded three cobalt ions forming a pillared host with three cobalt dithiolene cores, in which all sulfur and cobalt atoms were in one plane. The average Co–S bond distance of approximately 2.15 Å was in good agreement with the reported cobalt dithiolene species, implying that the cobalt dithiolene core retained its original redox activity after modification on the metal-organic cage (Supplementary Tables 1 and 2)[32,33]. Two triphenyl benzene groups, located on the top and the bottom of the triangular prism Co$_3$TPS$_2$, built a box-shaped cavity with a height of ~8.57 Å, wherein the negatively charged cobalt dithiolene cores were positioned at the midpoint of each edge with a Co···Co separation of ~18.66 Å and yielded a diameter of ~23.10 Å. This pillared cage containing two parallel aromatic planes provided an appropriate shape for encapsulating planar photosensitizer via aromatic stacking in a face-to-face way[35,36]. The amide groups which were located on the three triangular planes provided static, geometric, and functional properties to the cage, thereby enabling the metal-organic cage to attract electron donor via peripheral binding with a pocket-bound photosensitizer to form an integrated supramolecular assembly[32,37,38].

$^1$H NMR spectra displayed a single set of ligand related signals relative to a highly symmetrical complex, where a diffusion-ordered NMR spectrum confirmed the formation of a single species with a single diffusion coefficient of $8.6 \times 10^{-11}$ m$^2$ s$^{-1}$ and an estimated diameter of 23.40 Å based on the Stokes–Einstein equation consistent with the results determined by the crystal structure (Supplementary Fig. 4)[39,40]. The ESI-MS spectrum of the Co$_3$TPS$_2$ exhibited two intense peaks at $m/z = 625.6254$ and 949.9336 assigned to [Na$_{3-n}$Co$_3$(TPS)$_2$]$^{n-}$ ($n = 2$, 3) via a comparison with the simulation results based on natural isotopic abundances (Supplementary Fig. 3), demonstrating the high stability of the cage in DMF solution and the trivalent cobalt ions on Co$_3$TPS$_2$. There was the same outcome in the mixed solvent of EtOH/H$_2$O (3:2), showing the integrity of the triangular prism cage in a water-containing system.

Cyclic voltammetry of Co$_3$TPS$_2$ exhibited a suitable potential of −0.58 V (vs. Ag/AgCl) assignable to the couple Co(III)/Co(II), which is in good agreement with previously reported cobalt dithiolene species and falls well within the range of that for proton reduction in aqueous solution (Fig. 3a)[33,41]. The addition of trifluoroacetic acid triggered the emergence of a catalytic wave at approximately −0.75 and −1.20 V (Supplementary Figs. 13 and 14). Moreover, the catalytic response for the proton reduction permitted a linear dependence on Co$_3$TPS$_2$ concentration with a half-wave potential of −0.64 V in the presence of

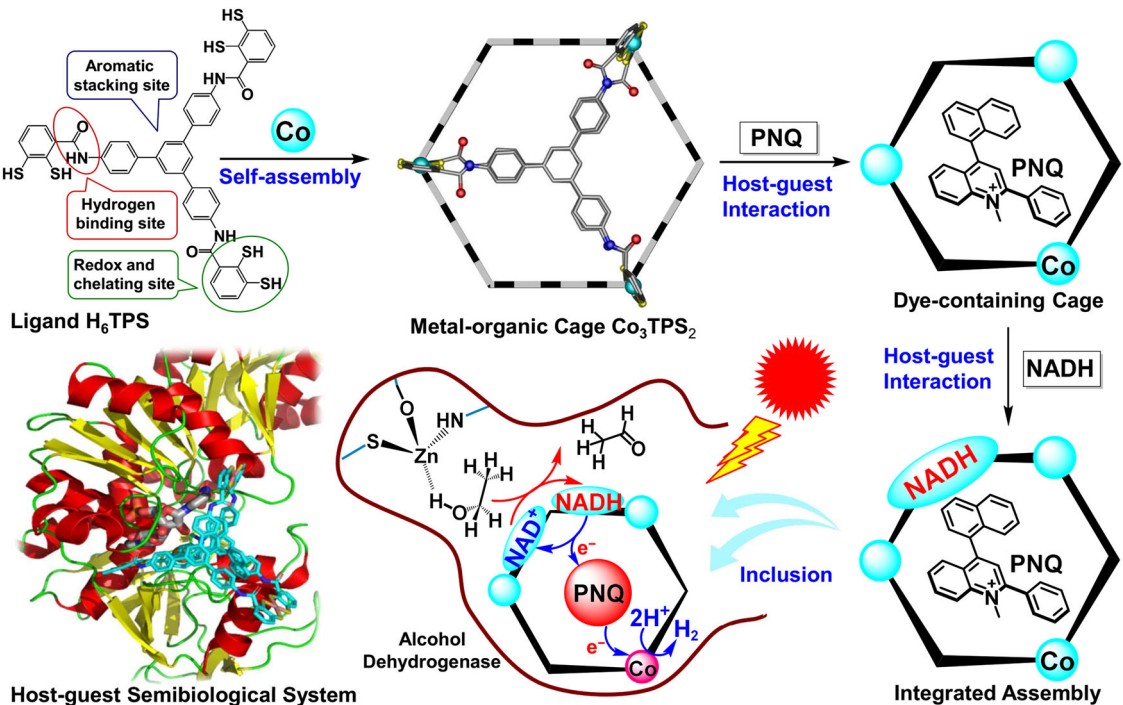

**Fig. 1 Schematic of the combination of artificial and natural enzymatic system.** Construction of the molecular triangular prism Co₃**TPS**₂, the dye-containing cage, the NADH-dye-cage ternary supramolecular system, and the host–guest semibiological system comprising metal-organic cage Co₃**TPS**₂ and natural enzyme ADH via non-covalent interactions, representing the assumed major binding conformation of the cage in the ADH enzymatic pocket from the docking study and the potential communication between artificial proton reduction and enzymatic alcohol dehydrogenation via the NAD⁺/NADH couple.

6.5 mM *p*-toluenesulfonic acid, therefore, the overpotential of Co₃**TPS**₂ could be estimated to approximately 0.16 V using the method of Evans[42,43], which is comparable to that of the reported cobalt dithiolene-containing catalysts (Supplementary Fig. 15)[34]. These electrochemical experiments indicated that the cobalt dithiolene functionalized Co₃**TPS**₂ still maintains intrinsic redox activity of the cobalt dithiolene species, and the noninnocent nature of the dithiolene moiety would allow protonation to take place at either the cobalt metal or sulfur for driving hydrogen production[33,41,44].

**Supramolecular photocatalysis in tandem with enzymes**. The artificial hosts featuring individual microenvironment are capable of limiting the supramolecular catalysis inside their pocket while coupling with the reactions outside[25]. Inspired by pioneering work[28], the positively charged dye **PNQ** that is capable of driving redox events with its moderate quencher NADH (Supplementary Fig. 9), possesses a size of 8.54 × 8.38 Å triangular plane which ideally matches the Co₃**TPS**₂ cavity. We anticipated that the π-electron-rich Co₃**TPS**₂ with multiple hydrogen bonding sites was able to co-encapsulate aromatic photosensitizer **PNQ** and electron donor NADH. The resulting close proximity between **PNQ** and NADH through host–guest approaches would be conducive to accelerating the photoinduced electron transfer from NADH moieties to the excited state of photosensitizer, giving a long-lived reduced photosensitizer to further reduce the cobalt dithiolene cores on the metal-organic cage for proton reduction and hydrogen evolution[45].

Photocatalytic attempts for proton reduction half-reaction using supramolecular catalyst Co₃**TPS**₂ (20.0 μM), photosensitizer **PNQ** (0.5 mM), and electron donor NADH (2.0 mM) was first explored in EtOH/H₂O (3:2) solution upon the irradiation of

300 W Xe lamp, resulting in an average of 40 μL hydrogen being produced after 4 h under optimized conditions (Supplementary Figs. 22–25 and Table 3, entry 1). We noticed that the photocatalysis driven by non-noble homogeneous catalyst Co₃**TPS**₂ exhibited similar catalytic conversion to that of noble metal catalytic system with 18% consumption of electron donors[28]. To the best of our knowledge, this is one of the few homogeneous photocatalytic systems that can use NADH as a direct electron donor to achieve artificial proton reduction[4], and this supramolecular catalytic system provides a distinctive approach to graft enzymatic system for light-trapped fuel and chemical conversion in a mild condition.

In order to achieve both photocatalytic hydrogen production and alcohol dehydrogenation at the same location, ADH (10 U mL⁻¹) and NAD⁺ (2.0 mM) were employed to replace NADH (2.0 mM) for the photocatalysis in EtOH/H₂O (3:2) solution containing **PNQ** (0.5 mM) and Co₃**TPS**₂ (20.0 μM). Noteworthily, the simultaneously generated hydrogen, and aldehyde were in a similar amount after 12 h, producing 147 μL hydrogen with a turnover number (TON) of 550 with respect to ADH, and 8.38 μmol aldehyde with a TON 700 with respect to ADH (Fig. 3b). This result demonstrated that the hybrid system consisting of metal-organic cages and natural enzymes could perfectly synergistically catalyze the NADH-modified hydrogen evolution and the NAD⁺-mediated alcohol dehydrogenation. The higher production of aldehyde than that of hydrogen indicated that the coenzyme NADH played an important role in storing protons and electrons, which eliminated the demand for transferring protons and electrons immediately, endowing the catalytic cycle with redundancy reminiscent of natural photosynthesis (Fig. 3b and Supplementary Fig. 32). In fact, the reaction could be extended up to 42 h and produced an average yield of 232 μL hydrogen with a TON of 875 with respect to ADH, exhibiting a

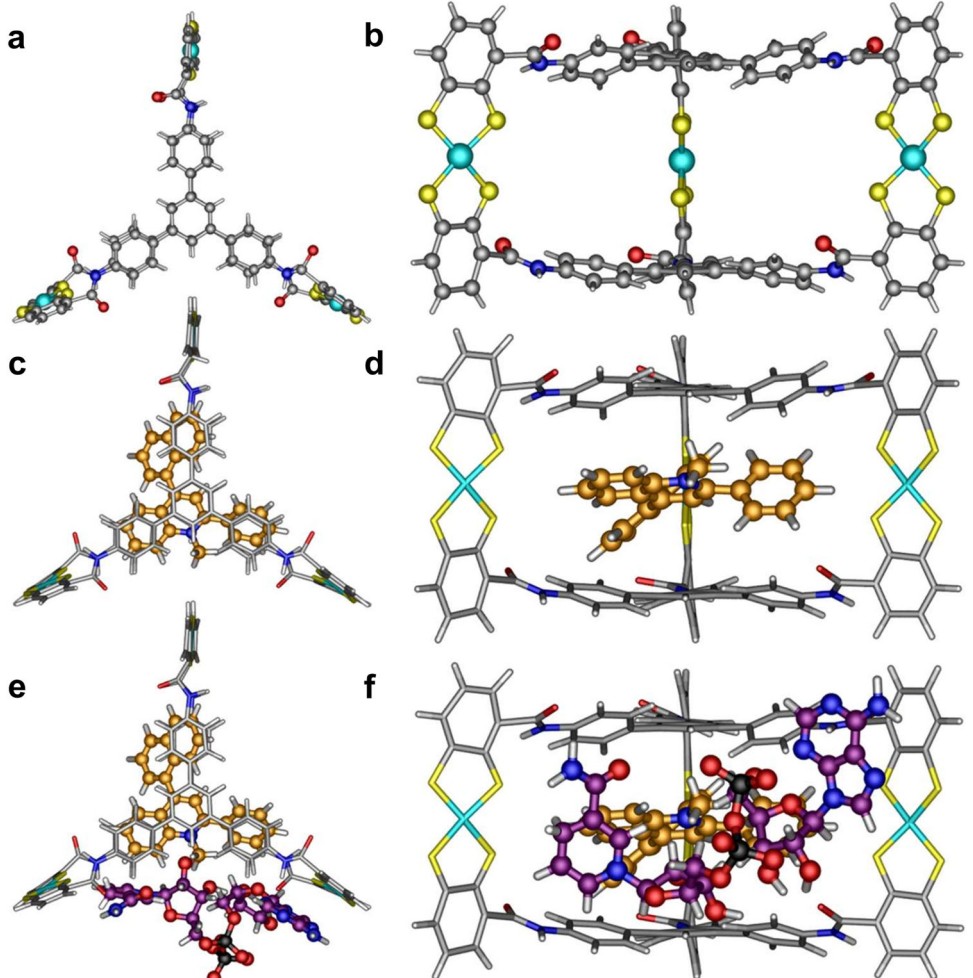

**Fig. 2 Structures of the artificial host and the host–guest species. a** Crystal structure of the triangular prism Co$_3$**TPS**$_2$ in top view, showing the large conjugate plane in the ligand H$_6$**TPS**. **b** Crystal structure of the triangular prism Co$_3$**TPS**$_2$ in main view, showing the coordination geometry of the cobalt dithiolene cores. Co cyan, S yellow, N blue, O red, C gray, and H white. **c** Top view and **d** main view of the theoretical docking study optimized model of Co$_3$**TPS**$_2$ ⊃ **PNQ**, showing the location of the **PNQ** in the center of Co$_3$**TPS**$_2$ cavity. **e** Top view and **f** main view of theoretical docking study optimized model of Co$_3$**TPS**$_2$ trapping both **PNQ** and NADH, showing the close proximity between the redox catalyst, photosensitizer and electron donor.

comparable catalytic activity to those reported similar supramolecular catalytic systems[32,46,47], while avoiding the use of sacrificial electron donors (Table 1, entry 1). However, the hydrogen production of the enzyme-free photocatalytic system no longer grew with the prolonging of time under the same reaction conditions (Fig. 3c). Control experiments revealed that the absence of any of these individual components led to a failure in hydrogen production (Table 1, entries 2–5), and the artificial system did not function well in the absence of light (Table 1, entry 6). Interestingly, the use of mononuclear compound Co**BDT**$_2$ (60.0 μM, ensuring the same concentration of cobalt ions; where **BDT** = 1,2-benzene-dithiolate)[33], which resembles a corner of the metal-organic cage Co$_3$**TPS**$_2$, yielded only 7 μL hydrogen following a shorter life of 12 h under same reaction conditions (Fig. 3b), despite the fact that the redox potential (−0.56 V vs. Ag/AgCl) of the electroactive Co**BDT**$_2$ is identical to that of Co$_3$**TPS**$_2$ (Fig. 3a and Supplementary Figs. 16, 17). Notably, the presence of Co**BDT**$_2$ also dramatically inhibited the production of aldehyde (Fig. 3b and Supplementary Fig. 31), which might be due to the inert binding of Co**BDT**$_2$ to ADH (Supplementary Fig. 21). These results demonstrated the ability of supramolecular catalysts to synergistically catalyze with natural enzymes in an efficient and compatible way, achieving a redox-neutral catalysis different from

**Table 1 Supramolecular semibiological system for light-driven alcohol splitting.**

| Entry | Catalyst | PNQ (mM) | NAD$^+$ (mM) | ADH (U mL$^{-1}$) | H$_2$ (μL) |
|---|---|---|---|---|---|
| 1 | Co$_3$**TPS**$_2$ | 0.5 | 2.0 | 10 | 232 |
| 2 | – | 0.5 | 2.0 | 10 | 0 |
| 3 | Co$_3$**TPS**$_2$ | – | 2.0 | 10 | 0 |
| 4 | Co$_3$**TPS**$_2$ | 0.5 | – | 10 | 0 |
| 5 | Co$_3$**TPS**$_2$ | 0.5 | 2.0 | – | 0 |
| 6[a] | Co$_3$**TPS**$_2$ | 0.5 | 2.0 | 10 | 0 |
| 7[b] | Co$_3$**TPS**$_2$ | 0.5 | 2.0 | 10 | 41 |

Reaction conditions: EtOH/H$_2$O (v:v = 3:2, pH 4.5), catalyst (20.0 μM), Xe 300 W, 42 h. The amount of hydrogen was determined by GC with an external standard method.
[a]In the absence of light.
[b]In the presence of inhibitor **DTQ** (0.1 M).

traditional ternary hydrogen evolution systems[33,41]. Moreover, an inhibition experiment was further carried out by adding a non-reactive species, 1,1-dimethyl-1,2,3,4-tetrahydroquinolinium salts (**DTQ**)[48], to suggest that the fine synergy of multiple electron transfer steps was realized by supramolecular host rather than a

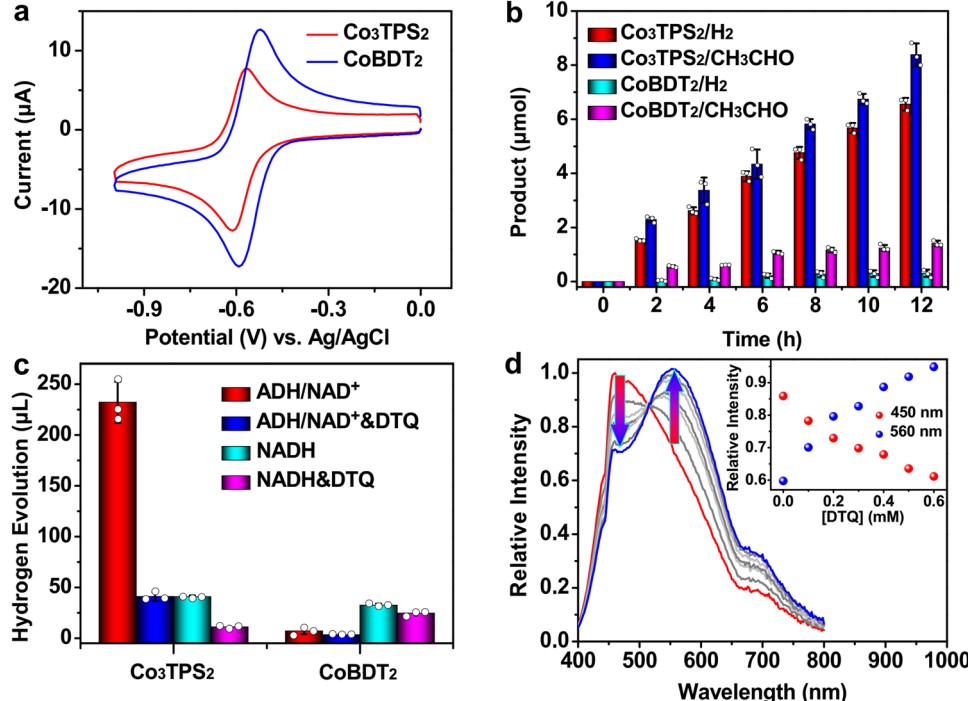

**Fig. 3 Catalytic properties of cobalt dithiolene-embedded catalysts. a** Cyclic voltammograms of the catalysts (0.1 mM) containing tetrabutylammonium hexafluorophosphate (TBAPF$_6$, 0.1 M) with a scan rate of 100 mV s$^{-1}$ in DMF. **b** Light-driven alcohol splitting with **PNQ** (0.5 mM), NAD$^+$ (2.0 mM), ADH (10 U mL$^{-1}$) and redox catalysts (20.0 μM for Co$_3$**TPS**$_2$ or 60.0 μM for Co**BDT**$_2$, ensuring the same concentration of cobalt ions) in EtOH/H$_2$O (3:2, pH 4.5) for 12 h. **c** Control experiment in EtOH/H$_2$O (3:2, pH 4.5) containing **PNQ** (0.5 mM), redox catalysts (20.0 μM for Co$_3$**TPS**$_2$ or 60.0 μM for Co**BDT**$_2$) and NADH (2.0 mM) or NAD$^+$ (2.0 mM) with ADH (10 U mL$^{-1}$), in the presence or absence of **DTQ** (0.1 M) within a 42 h period. Data points and error bars in **b** and **c** represent the mean ± s.d. of three independent experiments. **d** Luminescence spectra family of **PNQ** (0.1 mM), Co$_3$**TPS**$_2$ (0.1 mM) and NADH (0.1 mM) in EtOH/H$_2$O (3:2) upon the addition of **DTQ**. The inset shows the changes at 450 and 560 nm.

normal homogeneous manner. We speculated that the cation **DTQ**, which is similar in configuration to the photosensitizer **PNQ**, could compete to occupy the cavity of Co$_3$**TPS**$_2$, thereby blocking the orderly photoinduced electron transfer within metal-organic cage[32]. As expected, the addition of **DTQ** (0.1 M) into the optimal reaction system resulted in an effective quenching of the catalysis and gave only 17% hydrogen yield of the original system (Fig. 3c and Table 1, entry 7). These results indicated that the pre-organization effect of the host–guest system created an isolated catalytic microenvironment[16], and the formation of localized catalysis allowed the photocatalysis to not interfere with the reactions outside metal-organic cage, which was essential for coupling enzymatic reactions with benign compatibility[13].

Changing the ADH concentration showed that the initial rate of hydrogen production grew linearly with increasing ADH (from 2 to 10 U mL$^{-1}$), indicating that increasing ADH increased the more conversion of NAD$^+$ to NADH, which effectively supplied the protons and electrons needed for photocatalytic proton reduction (Fig. 4a and Supplementary Fig. 26). Increasing NAD$^+$ from 1.0 mM to 2.0 mM improved the amount of hydrogen production from 140 to 232 μL (Fig. 4b and Supplementary Fig. 27), indicating that a high coenzyme load promoted proton and electron capture during alcohol dehydrogenation. The formation of host–guest species, Co$_3$**TPS**$_2$ ⊃ **PNQ**, that serve as catalytic machines for proton reduction was verified by the linear increase seen in initial turnover frequency when Co$_3$**TPS**$_2$ was increased from 10.0 μM to 20.0 μM (Fig. 4c and Supplementary Fig. 28). When **PNQ** concentration was varied while maintaining the other parameters at a constant level (Fig. 4d and Supplementary Fig. 29), the impact of the amount of **PNQ** on overall catalysis was not significant compared to that of Co$_3$**TPS**$_2$,

implying that Co$_3$**TPS**$_2$ ⊃ **PNQ** played a dominant role in reaction acceleration, rather than **PNQ** itself[32]. Therefore, appropriately increasing catalyst content and reducing the photosensitizer amount, based on the original system, and performing photocatalysis with ADH (10 U mL$^{-1}$), NAD$^+$ (2.0 mM), **PNQ** (0.25 mM), and Co$_3$**TPS**$_2$ (40.0 μM) resulted in a higher yield amounting to 296 μL hydrogen with a TON of 1125 with respect to ADH, and 19.2 μmol aldehyde. In this case, the yield of H$_2$ was up to 132% (based on the concentration of coenzyme couple) and the conversion of NAD$^+$ to NADH based on the production of aldehyde was estimated to 192%. Significantly, all corresponding kinetic curves of supramolecular catalysis in collaboration with enzymes showed pseudo-zero-order kinetic behavior during the initial stages of reactions, and the initial rates of the reaction generally satisfied a Lineweaver–Burk plot with a concentration of NAD$^+$ or Co$_3$**TPS**$_2$ (Supplementary Fig. 30), further suggesting that both the metal-organic cage and coenzyme were located at ADH catalytic pocket for in situ catalysis, and this host–guest approach was able to transform the alcohol dehydrogenase into the alcohol lyases and created a continuously operating photosynthesis[26,27].

**Characterization of host–guest interactions.** To further understand the role of host–guest chemistry, the ESI-MS spectrum, isothermal titration calorimetry (ITC) essay, UV-Vis absorption spectra, circular dichroism (CD) spectra, fluorescence spectra, gel filtration chromatography, dynamic light scattering (DLS) analyses, and theoretical docking study were employed to provide insight into the host–guest interactions between functional components. The host–guest mode between Co$_3$**TPS**$_2$ and **PNQ**

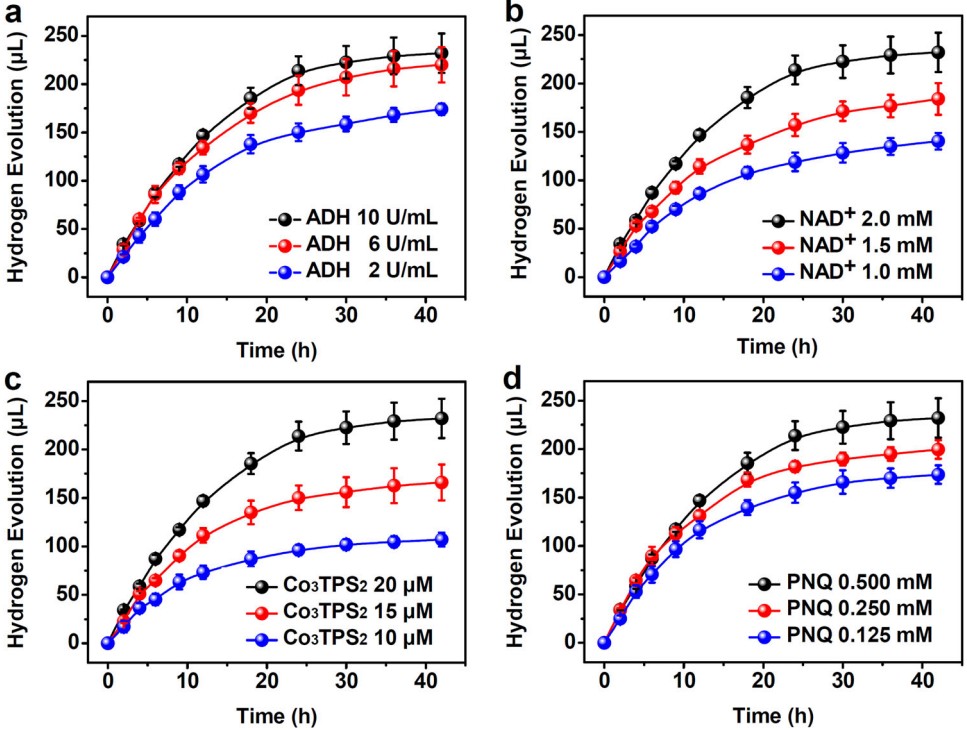

**Fig. 4 Kinetics of the photocatalytic hydrogen evolution from alcohol dehydrogenation.** Hydrogen evolution in EtOH/H$_2$O (3:2, pH 4.5) **a** as a function of the concentration of ADH with NAD$^+$ (2.0 mM), Co$_3$**TPS**$_2$ (20.0 μM) and **PNQ** (0.5 mM) remaining fixed; **b** as a function of the concentration of NAD$^+$ with ADH (10 U mL$^{-1}$), Co$_3$**TPS**$_2$ (20.0 μM) and **PNQ** (0.5 mM) remaining fixed; **c** as a function of the concentration of Co$_3$**TPS**$_2$ with ADH (10 U mL$^{-1}$), NAD$^+$ (2.0 mM) and 0.5 mM **PNQ** remaining fixed; **d** as a function of the concentration of **PNQ** with ADH (10 U mL$^{-1}$), NAD$^+$ (2.0 mM) and Co$_3$**TPS**$_2$ (20.0 μM) remaining fixed. Data points and error bars represent the mean ± s.d. of three independent experiments in all figures.

was first identified. Upon addition of **PNQ** (1.0 mM) into the solution of Co$_3$**TPS**$_2$ (1.0 mM), a sharp peak corresponding to [Co$_3$(**TPS**)$_2$·**PNQ**]$^{2-}$ at $m/z = 1111.5117$ was observed in ESI-MS spectrum (Supplementary Fig. 3), suggesting that **PNQ** could be included into Co$_3$**TPS**$_2$ to form 1:1 host–guest species[29]. ITC assay of Co$_3$**TPS**$_2$ upon the addition of **PNQ** gave a disassociation constant $K_{d1}$ measuring 6.99 μM, wherein the large change in Gibbs free energy ($\Delta G$) was calculated as $-29.43$ kJ mol$^{-1}$, showing the considerable affinity between the host Co$_3$**TPS**$_2$ and the guest **PNQ** molecule (Fig. 5a and Supplementary Fig. 18)[49].

Subsequently, the integration of Co$_3$**TPS**$_2$, **PNQ**, and NADH into one working module was explored. ITC measurment upon addition of coenzyme NADH into the solution containing both Co$_3$**TPS**$_2$ and **PNQ** yielded a $K_{d2}$ of 33.32 μM, and a $\Delta G$ of $-25.56$ kJ mol$^{-1}$ (Fig. 5a and Supplementary Fig. 19), suggesting the formation of an integrated supramolecular assembly in solution with a potential 1:1:1 stoichiometry[26,50]. This integrated binding was further supported by UV-Vis absorption and fluorescence spectra titration. The addition of NADH (0.1 mM) into a EtOH/H$_2$O (3:2) solution containing both Co$_3$**TPS**$_2$ (0.1 mM) and **PNQ** (0.1 mM) resulted in several isosbestic points in the absorption band fitting well with the 1:1 binding model at 660 nm (Fig. 5b and Supplementary Fig. 5)[51], and caused obvious emission quenching at 560 nm along with the emergence of a new blue-shifted peak at 450 nm agreeing with the non-linear Hill plot (Fig. 5c)[52,53], further confirming that recognition and assembly actually occurred when the coenzyme NADH encountered in a Co$_3$**TPS**$_2$/**PNQ** microenvironment. Although further forming multinary assembly with more NADH molecules was not observed, this possibility could not be ruled out.

Docking calculations suggested that the aromatic plane of **PNQ**, which fell on the center of the host cavity due to aromatic stacking interactions[54], highly overlapped the large conjugate plane of Co$_3$**TPS**$_2$ (Fig. 2c, d and Supplementary Fig. 34). NADH was in close proximity to the window of Co$_3$**TPS**$_2$ due to multiple hydrogen bonds yielding a calculated $\Delta G$ of $-24.58$ kJ mol$^{-1}$, a value which was in line with that estimated by the ITC test. This result suggested that spontaneous molecular behavior ensured a short distance and allowed direct communication between the nicotinamide moiety of electron donor NADH and photosensitizer **PNQ**, providing convenience for artificial catalytic system and natural enzyme to form integrated assembly (Fig. 2e, f and Supplementary Fig. 35)[50].

Importantly, the addition of **DTQ** into the Co$_3$**TPS**$_2$/**PNQ**/NADH system resulted in an emission recovery at 560 nm with a decrease in signal intensity at 450 nm (Fig. 3d), reflecting that the formation of supramolecular host–guest assembly led to the drastic retrenchment in terms of the distance between functional components, and the passivation behavior of the photocatalysis after adding inhibitor **DTQ** could be attributed to the destruction of the Co$_3$**TPS**$_2$/**PNQ**/NADH host–guest architecture through **DTQ**'s occupation in the cavity of Co$_3$**TPS**$_2$. The close proximity of NADH to **PNQ** enabled the completion of a pseudo-intramolecular electron transfer process from NADH via **PNQ** to the cobalt dithiolene moieties on the cage at a rate faster than the diffusion of NADH[29], which was intuitively validated via electron paramagnetic resonance spectra (Supplementary Fig. 11). Control experiments showed that when the addition of NADH (0.1 mM) into a EtOH/H$_2$O (3:2) solution containing both Co**BDT**$_2$ (0.3 mM) and **PNQ** (0.1 mM) triggered a quenching following linear Stern-Volmer fitting (Supplementary Fig. 10)[33].

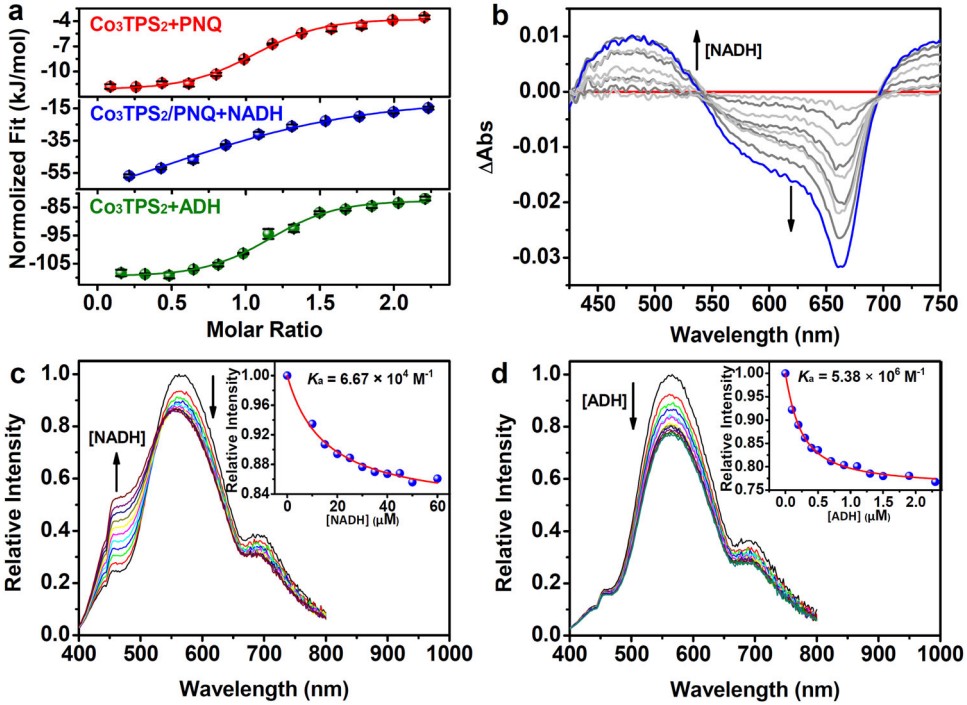

**Fig. 5 Characterization of host–guest interactions between Co₃TPS₂ and PNQ or NADH. a** Isothermal titration calorimetry tests of Co₃TPS₂ upon addition of **PNQ** (red) and ADH (green), and of the mixture of Co₃TPS₂ and **PNQ** upon addition of NADH (blue). Error bars are the calculated standard error from the curve fit. **b** UV-Vis absorption difference spectra of **PNQ** (0.1 mM) and Co₃TPS₂ (0.1 mM) in EtOH/H₂O (3:2) upon the addition of NADH. **c** Luminescence spectra family of **PNQ** (0.1 mM) and Co₃TPS₂ (0.1 mM) in EtOH/H₂O (3:2) upon the addition of NADH. Inset: Hill plot of the fluorescence intensity with a non-linear fitting at 560 nm (1:1 binding model). **d** Luminescence spectra family of **PNQ** (0.1 mM) and Co₃TPS₂ (0.1 mM) in EtOH/H₂O (3:2) upon the addition of ADH. Inset: Hill plot of the fluorescence intensity with a non-linear fitting at 560 nm (1:1 binding model).

This intermolecular collision behavior indicated that the catalysis in Co**BDT**₂/**PNQ**/NADH catalytic system was a normal homogeneous reaction different from the Co₃TPS₂/**PNQ**/NADH supramolecular system. These results indicated that the metal-organic host Co₃TPS₂ was able to integrate artificial catalytic components through host–guest interactions, which was beneficial to constraining an effective proton reduction inside the supramolecular host, and the formation of regional cooperation and division in catalysis was beneficial to joining outer enzymatic reactions for a redox-neutral artificial photosynthesis.

The host–guest relationship between artificial catalyst and natural enzyme was further considered. Both UV-Vis and CD spectra showed the characteristic peaks attributable to ADH were basically maintained after adding the cage Co₃TPS₂ (Supplementary Fig. 6). The intensity of Co₃TPS₂ (10.0 μM) at 300 nm decreased linearly when treated with a small amount of ADH (total 0.65 nM) (Supplementary Fig. 7), and the related difference spectra of UV-Vis absorption revealed a significant spectra changes centered at 268 nm from an initial linear growth to almost unchanged (Supplementary Fig. 8), which could be interpreted as the steady conversion of ADH to ADH ⊃ Co₃TPS₂ complex[26,55,56]. Luminescence titration of dye-containing cage (0.1 mM Co₃TPS₂ and **PNQ**) upon the addition of ADH in EtOH/H₂O (3:2) exhibited significant quenching of luminescent intensity at 560 nm, and the titration curve coinciding with the non-linear Hill plot (Fig. 5d). The observed quenching behavior was probably attributed to the capture of the dye-containing cage by ADH catalytic pocket via non-covalent interactions similar to the reported artificial host–guest system[29]. Elaborate calculations of the thermodynamics of Co₃TPS₂ binding to ADH were

conducted using ITC evaluation (Fig. 5a and Supplementary Fig. 20), which yielded a disassociation constant $K_{d3}$ of 0.23 μM accompanied with a large $\Delta G$ of $-37.90$ kJ mol⁻¹, reflecting the high affinity between the cage Co₃TPS₂ and enzyme ADH supported by supramolecular interactions[26,57]. Gel filtration chromatography showed that the sample of ADH with Co₃TPS₂ emerged a new peak with a shorter retention time than that of ADH, implying that Co₃TPS₂ bound to ADH giving a larger hydrodynamic radius (Supplementary Fig. 12a). The DLS measurement of ADH exhibited a sharp size-distribution peak and presented an average hydrodynamic radius of ~6.9 nm, while an average hydrodynamic radius of ~7.3 nm after adding the cage Co₃TPS₂ (Supplementary Fig. 12b). Theoretical docking study revealed that the dye-containing cage was capable of binding to the ADH catalytic pocket and burying its active cobalt dithiolene moiety into ADH to form a working module (Supplementary Fig. 36). The random binding model manifested that the cage binding to the ADH catalytic pocket was the major binding conformation with a higher binding energy, which might benefit from the suitable opening and abundant non-covalent interaction site of the enzymatic pocket (Supplementary Fig. 37). In addition, the coenzyme NADH was locked in the enzymatic pocket, serving as a communicator to couple artificial and natural enzymes. The generated NAD⁺ could directly participate in the enzymatic alcohol dehydrogenation, eliminating the diffusion process of coenzyme in the bulk solution and leading to a redox-neutral photosynthesis system (Supplementary Fig. 37)[26,27,58]. These findings indicated that the careful orchestration of a dye-containing metal-organic cage into the enzymes catalytic pocket allowed the in situ communication between the abiotic

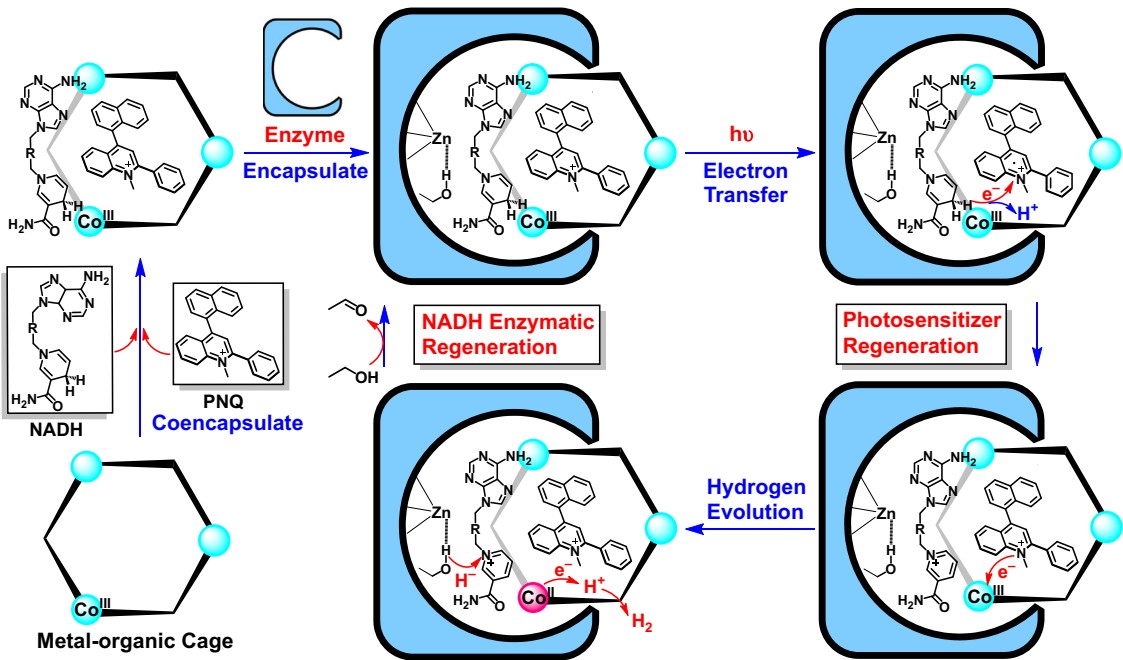

**Fig. 6 Proposed mechanisms for abiotic–biotic coupled system in photocatalytic alcohol splitting.** Schematic of semibiological supramolecular system for solar alcohol splitting showed that the coenzyme locked in the ADH catalytic pocket joined artificial photocatalysis with enzymatic reactions in situ.

and biotic systems through coenzymes. In this situation, the transport of electrons and matters during photosynthesis was strengthened in a closed loop of electrons and protons, which deserved better synthetic sequences with finer synergy.

**Proposed mechanisms of catalysis**. This light-driven supramolecular host–guest semibiological system was well modified for the simultaneous combination of dye-mediated artificial catalysis with a biotic catalysis at the pocket of ADH enzyme, which allowed the NADH-mediated photocatalytic proton reduction in tandem with $NAD^+$-mediated enzymatic alcohol dehydrogenation in situ (Fig. 6). The well-designed artificial metal-organic host, which possesses cobalt dithiolene catalytic cores on the skeleton and stacking interaction sites in the hydrophobic pocket and multiple hydrogen bonding sites at the opening windows, could simultaneously trap a photosensitizer and coenzyme to restraint the photoinduced hydrogen production from alcohol dehydrogenation inside its pocket. The close proximity between electron donors, electron acceptors, and catalytic cores guided a pseudo-intramolecular electron transfer from NADH moieties to the excited photosensitizer, and then to further reduce the cobalt dithiolene core on the metal-organic cage for proton reduction. With the intervention of enzymes, the artificial host coerced the entire photocatalytic proton reduction half-reaction into the ADH catalytic pocket and formed a superstructure containing both artificial and natural catalytic system. In this case, the transfer of matters and energy between the abiotic and biotic catalytic modules could be enhanced through close contact, which avoided the diffusion of coenzymes and the expense of sacrificial reagents. The coenzyme $NADH/NAD^+$ regenerated in redox catalysis was used as a direct electron/proton supply in situ for the next catalytic cycle, enabling a smooth switching between the abiotic photocatalysis and established enzymatic reactions. In the meanwhile, the inherent confined effects provided by both artificial host and enzyme allowed the formation of relatively independent catalytic processes, eliminating inherent mutual interference between the abiotic and biotic systems and promoting the formation of a continuously running redox-neutral photosynthesis system.

## Discussion

In summary, a redox-active metal-organic cage $Co_3TPS_2$ as a hydrogenase analog was embedded into the catalytic pocket of natural enzyme ADH through supramolecular interactions for solar alcohol splitting. The abundant non-covalent interaction sites in the artificial host allowed it to form an integrated host–guest species with a photosensitizer and an electron donor, constraining the photocatalytic hydrogen production from alcohol dehydrogenation inside the supramolecular host. The direct proton and electron delivery at close range between the two redox catalytic cycles provided positive feedback to the alcohol dehydrogenation processes. The attempt to associate artificial enzyme with non-photoactive natural enzyme in a host–guest approach achieved the optimized allocation of matter and energy by forming regional cooperation and division and insured electron transfer more efficient and controllable, illuminating the superiority of this supramolecular host–guest approach for redox-neutral artificial photosynthesis and providing a potential way to reduce carbon dioxide and even nitrogen.

## Methods

**General methods and materials**. [1]H NMR dates were collected on a Bruker 400M spectrometer with chemical shifts reported as ppm (in DMSO-$d_6$ or $CDCl_3$, TMS as internal standard). Elemental analyses of C, H, and N were performed on a Vario EL III elemental analyzer. ESI mass spectra dates were collected on a HPLC-Q-Tof MS spectrometer using acetonitrile as mobile phase. ITC essays were performed on a Nano ITC (TA Instruments Inc., Waters LLC). UV-Vis spectra were performed on a HP 8453 spectrometer. Fluorescent spectra were performed on Edinburgh FS-1000. CD spectra were measured on a JASCO J-810 spectropolarimeter. DLS measurements were performed on Malvern Zetasizer Nano ZS90 analyzer. EPR spectra were measured on a Bruker E500 spectrometer. Electrochemical measurements were carried out under Ar at room temperature and performed on a ZAHNER ENNIUM electrochemical workstation with a conventional three-electrode system with an Ag/AgCl electrode as a reference electrode, a platinum silk with 0.5 mM diameter as a counter electrode, and glassy carbon electrode as a working electrode. Gel filtration chromatography was performed on AKTA purifier 100 using a Sephadex G-75 gel sieving column, UV-Vis detector (detection wavelength 280 nm), 1×PBS as mobile phase. Unless stated otherwise, all chemicals were of reagent grade quality obtained from commercial sources, biomaterial ADH from Saccharomyces cerevisiae was purchased from Sigma-Aldrich. Solvents were dried by standard methods and freshly distilled prior to use. All synthesis operations were carried out under an atmosphere of dry argon using Schlenk and vacuum techniques. The photosensitizer **PNQ** and catalyst

$Co\mathbf{BDT}_2$ were synthesized according to the reported procedures by Fukuzumi[28] and Eisenberg[33], respectively. Ligand $H_6\mathbf{TPS}$ was synthesized similar to the reported procedures[32,59,60].

**Synthesis of $H_6\mathbf{TPS}$ precursor.** Freshly prepared 2,3-bis(isopropylthio)benzoyl chloride was dissolved in THF (20 mL) and this solution was added to a solution of 4, 4′, 4″-triaminotriphenyl-benzene (0.5 g, 1.30 mmol) and $NEt_3$ (2.0 mmol) in THF (40 mL) at 0 °C. Then, the reaction mixture was stirred for 12 h at ambient temperature. Subsequently, insoluble material was removed by filtration and the solvent was removed from the filtrate under vacuum. The pure product was obtained after washing with diethyl ether. Yield: 1.4 g, 92%. $^1H$ NMR ($CDCl_3$, 400 MHz, ppm): δ 9.07 (s, 3H; NH), 7.80–7.64 (m, 18H; ArH), 7.42–7.35 (m, 6H; ArH), 7.11 (d, $J$ = 8.8 Hz, 6H; ArH), 3.55-3.41 (m, 6H; $(CH_2)_2$), 1.41 (d, $J$ = 6.7 Hz, 18H; $CH_3$), 1.24 (d, $J$ = 6.7 Hz, 18H; $CH_3$). $^{13}C$ NMR ($CDCl_3$, 101 MHz, ppm): δ 166.1, 146.0, 142.0, 141.8, 137.6, 137.2, 129.2, 128.9, 128.5, 128.0, 126.4, 124.5, 120.3, 41.4, 36.3, 23.1, 22.7. ESI-MS calcd for $C_{63}H_{69}N_3O_3S_6$: 1107.37, found 1108.37 $[M+H]^+$, 1130.36 $[M+Na]^+$. Elemental analysis calcd for $C_{63}H_{69}N_3O_3S_6$: H, 6.27; C, 68.25; N, 3.79%; found: H, 6.34; C, 67.66; N, 3.75%.

**Preparation of $Co_3\mathbf{TPS}_2$.** Freshly distilled THF (20 mL) was added to a mixture of precursor (387.6 mg, 0.35 mmol), sodium (181.1 mg, 7.85 mmol), and naphthalene (336.5 mg, 2.60 mmol), and the reaction was stirred for 12 h at 25 °C. Subsequently, methanol (5.0 mL) was added to remove unreacted sodium and reaction solvents were removed under vacuum. The solid residue was dissolved in degassed water and the resulting solution was washed three times with degassed diethyl ether ($3 \times 20$ mL). The aqueous solution was filtered and acidified with HCl (37%) to give a white precipitate, $H_6\mathbf{TPS}$, which was used directly to stir with NaOH (85.6 mg, 2.14 mmol) and $Co(BF_4)_2\cdot6H_2O$ (183.9 mg, 0.54 mmol) in a DMF solution (30 mL) for 12 h. Then, $NEt_4Cl$ (89.5 mg, 0.54 mmol) was added to this solution and stir for another 4 h. Then, the solution was poured into diethyl ether for a dark blue precipitate. The solid was collected and redissolved in DMF, and the dark blue crystals of $Co_3\mathbf{TPS}_2$ suitable for single-crystal X-ray diffraction were obtained by diffusing diethyl ether into the DMF solution, yield: 51%. Elemental analysis calcd for $Co_3(C_{45}H_{27}-N_3O_3S_6)_2\cdot(NC_8H_{20})_3\cdot(C_3H_7NO)$: H, 5.21; C, 60.03; N, 5.98%; found: H, 5.31; C, 59.96; N, 6.21%. ESI-MS: $m/z$ = 625.6254 $[Co_3(\mathbf{TPS})_2]^{3-}$, 949.9336 $[NaCo_3(\mathbf{TPS})_2]^{2-}$.

**X-ray crystallography.** The intensities of the $Co_3\mathbf{TPS}_2$ were collected at 180(2) K on a Bruker SMART APEX CCD diffractometer equipped with graphite mono-chromated Mo-Kα ($\lambda$ = 0.71073 Å) radiation source and the data were acquired using the SMART and SAINT programs[61,62]. The structure was solved by direct methods and refined on $F^2$ by full-matrix least-squares methods with SHELXTL version 5.1 software[63]. In the structural refinement of $Co_3\mathbf{TPS}_2$, all the non-hydrogen atoms were refined anisotropically. Hydrogen atoms within the ligand backbones, a DMF molecule, and three $NEt_4^+$ cations were fixed geometrically at calculated distances and allowed to ride on the parent non-hydrogen atoms. To assist the stability of refinements, two amide groups on the ligands, a DMF molecule, and two $Et_4N^+$ cations were limited to the desired position with rational thermal parameters by several restrains. One methyl on an $Et_4N^+$ cation, and a carbon atom and nitrogen atom on the DMF were disordered into two parts with s.o.f of each part being refined using free variables. The thermal parameters on adjacent atoms in all $Et_4N^+$ cations, a DMF molecule, and some parts of ligands were restrained to be similar. In addition, the SQUEEZE subroutine in PLATON was used for refinements[64].

Crystal data of $Co_3\mathbf{TPS}_2$: $Co_3(C_{45}H_{27}N_3O_3S_6)_2\cdot3NC_8H_{20}\cdot C_3H_7NO\cdot4.5H_2O$, $M$ = 2421.81, Triclinic, space group $P-1$, $a$ = 12.8740(12), $b$ = 27.316(3), $c$ = 27.453(4) Å, $\alpha$ = 119.649(3), $\beta$ = 98.301(6), $\gamma$ = 94.063(4), $V$ = 8187.8(15) Å$^3$, $Z$ = 2, $Dc$ = 0.982 g cm$^{-3}$, $\mu$(Mo-Kα) = 0.499 mm$^{-1}$, $T$ = 180(2) K. 28,559 unique reflections [$R_{int}$ = 0.1044]. Final $R_1$ [with $I > 2\sigma(I)$] =0.1074, $wR_2$ (all data) = 0.2388 for the data collected. CCDC number 2042990.

**General methods for theoretical 'docking study'.** Docking calculations were performed with the AutoDock program 4.2. The $Co_3\mathbf{TPS}_2$, **PNQ**, and NADH were downloaded from the CCDC database. The structure of enzyme alcohol dehydrogenase was downloaded from the Protein Data Bank (PDB) database (PDB code: 5ENV). The cage $Co_3\mathbf{TPS}_2$ was used to perform the docking calculation after energy minimization. The models of the enzyme were refined by removing hydrogen atoms. Polar hydrogens were then added, followed by the assignment of Kollman charges, fragmental volumes, and atomic solvation parameters to adhesive by means of AutoDock Tools. For the ligand, the molecule was refined by removing and subsequently adding hydrogen atoms in a similar manner to that for adhesive. Next, Gasteiger partial charges were assigned to the ligands, and nonpolar hydrogens were merged. All torsions were allowed to rotate during docking. The Lamarckian genetic algorithm was used to determine the appropriate binding positions, orientations, and conformations of the ligands. Default parameters were used, except for the number of generations which was set to 300. The blind docking strategy was used with a 50 Å × 78 Å × 114 Å grid box which ensured sufficient spaced to cover the entire surface of the enzyme. The Lamarckian genetic algorithm was chosen with default

parameters except for the number of generations, which was set to 100 for more accurate docking results. The best docking mode of the host–guest complex was chosen based on the binding energy score, clustering, and chemical reasonableness.

**General methods for photocatalysis.** General method for photocatalytic proton reduction. Varying amounts of the catalyst and **PNQ** were added into an EtOH/$H_2O$ solution (v:v = 3:2, pH 4.5, 5.0 mL) containing NADH with a magnetic stir bar. The flask was sealed with a septum and protected from air by Ar. The samples were irradiated by a 300 W Xenon lamp. The reaction was maintained at 25 °C by using a water filter to absorb heat. The general method for photocatalytic alcohol splitting. Varying amounts of the catalyst, ADH and **PNQ** were added into an EtOH/$H_2O$ solution (v:v = 3:2, pH 4.5, 5.0 mL) containing $NAD^+$ with a magnetic stir bar. The flask was sealed with a septum and protected from air by Ar. The samples were irradiated by a 300 W Xenon lamp. The reaction was maintained at 25 °C by using a water filter to absorb heat. The generated hydrogen was characterized by GC 7890T instrument analysis using a 5 Å molecular sieve column, thermal conductivity detector, and argon used as carrier gas. The amount of hydrogen generated was determined by external standard method[65]. The generated aldehyde was characterized by an Agilent 6890N GC system using a FFAP capillary column, flame ionization detector, and nitrogen used as carrier gas. The amount of aldehyde generated was determined by external standard method[17,66].

## Data availability
The X-ray crystallographic coordinates for the structures reported in this article have been deposited at the Cambridge Crystallographic Data Centre under the deposition numbers CCDC 2042990. The data can be obtained free of charge from the Cambridge Crystallographic Data Centre via http://www.ccdc.cam.ac.uk/data_request/cif. Enzyme ADH structure data with the accession code 5ENV was downloaded from the PDB database via https://www.rcsb.org/. All other data supporting the findings of this study are available within the article and its Supplementary Information files or from the corresponding author upon request. A reporting summary for this article is available. Source data are provided with this paper.

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

## Acknowledgements

This work was supported by the National Natural Science Foundation of China (Nos. 21820102001, 21861132004, and 21890381) and the Fundamental Research Funds for the Central Universities (DUT20TD101).

## Author contributions

J.C., L.Z., C.H., and C.D. conceived the project and designed the experiments. J.C. carried out the main experiments, collected and interpreted the data. Y.L prepared the ligand. J.C., L.Z., and C.H. solved and refined the X-ray single-crystal structures. L.Z., C.H., and C.D. contributed materials and analysis tools. J.C., L.Z., and C.D. co-wrote the paper. All authors discussed the results and commented on the manuscript.

## Competing interests

The authors declare no competing interests.

## Additional information

**Peer review information** *Nature Communications* thanks Michael Ward and other, anonmyous, reveiwers for their contributions to the peer reivew of this work. Peer review reports are available.

