## [Peer Review File · Nature Communications]

REVIEWER COMMENTS

Reviewer #1 (Remarks to the Author):

The manuscript entitled "A host-guest semibiological photosynthesis system: coupling artificial and natural enzymes for solar alcohol splitting" describes a novel combination of synthetic and enzymatic systems for photocatalytic hydrogen evolution via alcohol dehydrogenation. The authors appear to build on the success of their recent efforts (Nat. Commun. 2020, 11, 2903–2913) to enhance photocatalysis through biohybrid constructs in which a metallo-organic artificial enzyme can be regenerated by a biological enzyme to which it is bound. While this investigation offers interesting and promising insights on how to design supramolecular biohybrid assemblies for coupled catalysis, there are a number of major and minor issues with the manuscript that may inform the authors' and editors' revisions and decisions.

MAJOR ISSUES:

Throughout this work (and the previous Nature Communications paper by the same team), the authors use the terms "host-guest" and "matryoshka" to describe the binding of the metallo-organic cage "inside" the enzyme's catalytic domain. This internal binding is a central claim of the paper. In order to use these terms, the authors should prove that cage binds "inside" a protein cavity rather than "outside" on an external surface of the protein. What experimental evidence is there for this "inside" binding of the cage to the enzyme? I can only find support in the form of the authors' claim that "Co-TPS (10.0 μ M) at 300 nm decreased linearly when treated with ADH suggesting that the artificial catalyst was shielded by the cavity of ADH⁵⁵". However, I suspect this linear decrease (Figure S6) is simply attributable to dilution. If not, the authors need to substantiate the claim, because their citation of reference 55 does not explain or support it. Reference 55 does not discuss how a linear decrease in UV absorption supports endohedral binding (indeed, quite the opposite, the authors of that paper admit they cannot obtain binding data from their UV-Vis spectra on page S27 of their supplemental document) nor does the paper involve ADH or any protein binding whatsoever. It is not unlikely that the Co-TPS cage binds to the ADH protein via multiple non-specific surface interactions involving hydrogen bonding and charge-pairing interactions, which are likely strengthened by the presence of a large proportion of organic solvent in their experiments. This counter-hypothesis to the "inside" binding claim does not necessarily preclude the artificial enzyme from enhanced catalytic function by virtue its proxy to the enzyme's active site, and could also explain why Co-TPS interacts with the protein more strongly than Co-BDT (if that is the case – the Co-BDT-protein interaction is not discussed in the paper), since Co-BDT has far fewer charged and hydrogen bonding sites than Co-TPS.

The authors do offer a computational evidence for the "inside" docking in the form of a theoretical calculation, but this calculation is not compared energetically with non-specific "outside" docking interactions, and even these models (Fig. 29) appear to show the cage at the proteins surface, with only the NADH/NAD⁺ molecules in the catalytic domain.

Figure 1: The cartoon of the "artificial enzyme" in Fig. 1 and Fig. 6 is confusing; it doesn't show a representation of the prismatic Co-TPS cage; only the NADH molecules the cage allegedly "traps" – this drawing initially misleads the reader into thinking that the NADH molecules are the ligands of the cage bound to cobalt, which is not true. The authors should revise their schematics to more accurately reflect the structure of the artificial enzyme complex.

If NADH can be added to PNQ@Co-TPS to form a ternary assembly, could it not also add a second and third time to form quaternary and quinary assemblies too? This issue seems not to be discussed even while the cartoon in Fig. 1 seems to imply it. The ITC data seems to be fit to a 1:1 binding of NADH to PNQ@Co₃TPS₂; why doesn't NADH bind a second and third time to the other bay regions of the host? The citations 23, 49, and 50 on line 257 do not offer an obvious explanation.

Several of the supplementary figures are not mentioned or discussed at all in the main text nor the supplementary information file. These figures should be cited and discussed or else removed entirely if they are not worth talking about. (Figures S4 S5 S9 S13 S14, possible others)

Lines 188-190: "This result suggested that the proton reduction catalysis driven by Co-BDT interfered with the enzymatic reactions, since a stable supramolecular host-guest semibiological system could not be realized." How do the authors know a stable host-guest system was not realized for Co-BDT? There is no supporting data (e.g., ITC) to show that Co-BDT does not bind ADH. Furthermore, is it possible that binding "inside" by Co-BDT could be responsible for this inhibitory effect by blocking reactive site access for substrates?

Why are there no error bars in any of the quantitative data (Figs. 3b-c, 4, 5a, S14-S25)? Does it mean the experiments each performed only once? If so, why should readers trust the data from one-time experiments that were not repeated?

MINOR ISSUES:

The authors start several sentences with "And" (lines 15, 41, 112, 279, 326) which is awkward and best avoided.

Line 98: Why do the authors cite reference 41 when they claim their cage is appropriately shaped for binding PNQ via aromatic stacking? Reference 41 does not discuss PNQ binding, while there are many prior works by Makoto Fujita describing aromatic stacking in metallacages more generically. Reference 42 is also questionable in this regard as it discusses DNA intercalation.

Page 4: Should the authors refer to the CoS4 cores as CoS4⁻ cores to indicate the cobalt (III) ion and not imply a Cobalt (IV) oxidation state?

I recommend the authors refer to the cage as Co₃TPS₂ or Co₃(TPS)₂³⁻ instead of Co-TPS, since the latter may mislead the reader into thinking there is an equal portion of Co ions and TPS ligands.

Reference 39, change "Common." to "Commun." (Line 610)

Reference 43, change "Nelsonbc" to "Nelson"

The synthetic details in the SI are confusing. For example:

- In the synthesis of H6TPS, compound 5 is referred to by its IUPAC name on page 6, line 100, even though the IUPAC name is not mentioned earlier in the synthesis of compound 5. The authors should stick with "compound 5" or define compound 5 by the IUPAC name the first time it is used.
- SI Page 6, Line 113, what is "precursor"? Presumably it is H6TPS, but the authors should be specific about which reagents they are using in a procedure.
- Line 103, the "H6TPS precursor" is described as a "pure product" but then it is purified to afford "H6TPS ligand" in lines 112-123. Why do the authors call it a pure product if this purification step is required prior to adding cobalt?

Please specify which reagents are variable in the actual graphs of Fig. 4 so readers do not have to search the captions to find the information.

Figure 5a is difficult to read because it has 3 axes. Perhaps the plots can be fit on a single axis with insets to zoom in on the lower-amplitude curves.

Reviewer #2 (Remarks to the Author):

This is a well-done and interesting example of a supramolecular system which combines biotic and artificial components to effect simultaneously light-induced oxidation and reduction processes. The reduction (at the artificial Co/dithiolene redox site) is of two protons to H₂; the coupled oxidation is of ethanol to the aldehyde by an alcohol dehydrogenase enzyme. Thus, the overall result is light-induced conversion of ethanol to aldehyde + H₂, conceptually similar to photosynthetic water-splitting.

The coupling of the two redox cycles has been the focus of much recent work from this (and other groups) in recent years. The work reported here constitutes a very nice example which is a significant advance in this field and it deserves publication in Nature Comms. The work is meticulously thorough and the conclusions well-evidenced with all necessary control experiments in place; in particular the simultaneous binding of NADH and the sensitiser in the host cavity to give a 1:1:1 ternary complex, an essential prerequisite for this process to work, has been established by spectrophotometric and ITC titrations. Further, there is evidence that this 1:1:1 assembly interacts with the dehydrogenase enzyme active site which is also essential for efficient coupling of the two halves of the process.

I have just one question for the authors to consider. The reaction progress is monitored by formation of H₂, which is relatively easy to measure. Of course, this should be accompanied by formation of an equivalent amount of ethanal. If there is some way to demonstrate ethanal formation, and confirm that the quantity matches the H₂ production, that would be very nice. There are various possible methods (maybe IR, GC-MS, NMR...) but simplest might be a colorimetric assay like this:

<https://www.sigmaaldrich.com/catalog/product/sigma/mak140?lang=en&ion=GB#:~:text=The%20Colorimetric%20Aldehyde%20Assay%20Kit,the%20amount%20of%20aldehyde%20present.>

Subject to consideration of this possibility I therefore recommend acceptance of this interesting and well-done study.

Mike Ward

Reviewer #3 (Remarks to the Author):

The authors describe a host-guest system that couples the action of the enzyme alcohol dehydrogenase with a dithiolene-embedded MOF hosting an organic dye as a photosensitizer. This system yields H₂ through NADH-mediated hydrogen production, where NADH (and a proton) result from ethanol dehydrogenation (alcohol + NAD⁺ → aldehyde + NADH + H⁺). Hydrogen production is catalyzed by an "artificial enzyme" consisting of a cobalt dithiolene complex embedded in a MOF, which also contains the ADH enzyme and the photosensitizing dye. Overall, I like this approach that combines MOF chemistry, Biochemistry, coordination chemistry, and photocatalysis for solar fuels. I find the approach to be quite clever.

It took me many reads of the manuscript to understand the system in place here because the presentation of the system is confusing. One major issue around clarity is that the structure and composition of the cobalt catalyst is not made clear early in the paper. While I came to understand through reading further that the cobalt dithiolene sites in the MOF, described as Co-TPS, are catalytic sites for hydrogen evolution, it was difficult to figure out whether Co-TPS is a MOF node or a catalyst or both. It is not clear to me what the structure is that is shown as the "artificial enzyme" in Fig 1? Is that a different cobalt catalyst? Also, what TPS stands for was not defined. Fig. 1 overall is confusing. Does the alcohol dehydrogenase (red outline) encapsulate the PNQ dye and also the cobalt active site for H₂ production? What is the relationship between the MOF structure and the enzyme?

There are some statements in the manuscript that are not well supported. The CoS4 sites in the MOF are said to yield "outstanding redox activity with a particularly low overpotential." How is outstanding redox activity defined and measured? Electron transfer rates or some other measure? I do not see any overpotential measurement either or indication of how the overpotential here is referenced (with respect to proton reduction...under what conditions and using what reagents?) The authors report a CV of the Co-TPS (Fig. 3a) and relate it to cobalt dithiolene species but that is a reversible CV, not a catalytic CV, and no substrate is added to test whether this redox couple supports catalysis (assuming this is collected in aprotic solvent, as suggested by the electrolyte given, but solvent is not indicated here – if this is collected in H₂O/EtOH, then this redox couple is not likely involved in catalysis).

It is clear that this system works, but the activity is not too impressive, but also not made very clear. In Fig. 4, the authors the volume of H₂ produced over time and my quick comparison to other systems in the literature is that it on the order of what is typically seen in systems combining Ru(bpy)₃ with a molecular catalyst and ascorbic acid. However, the authors do not compare their activity to other systems, and do not give a turnover number or information such as initial rate. They give some information on initial rate in Figure S23-25 as a function of conditions but again these results are not compared to other systems in the literature. Also, critically, I do not find a turnover number reported anywhere in the manuscript or in the SI. Without a turnover number, it is not even verified that the system is truly catalytic (although I suspect it is based on the data shown, nevertheless a TON is needed). A TON for production of H₂ with respect to catalyst and photosensitizer both could be determined, assuming that each cobalt site available can be a catalytic site; TON with respect to ADH also could be reported. I realize that TON is tricky to define in such a system but it can't be ignored.

The proposed mechanism is quite rudimentary. That's not a problem as Fig. 6 is a good clarifying figure, but the work gives minimal real insight into mechanism. For example, the key fundamental question of whether photosensitizer quenching is reductive or oxidative is not even addressed. I also am concerned about the authors proposing Co(II) to be the active species in proton reduction (Table 1), which is not supported.

In summary, this manuscript is much too preliminary to be published. It presents a potentially interesting system but its activity is not well characterized and the approach and analysis is not clearly explained to the reader. The investigation of the mechanism neglects fundamental questions of how electron transfer occurs in the system. The electrochemical study looks only at a reversible couple likely irrelevant to catalysis. Finally, the activity is not placed into context of the field through comparison to other relevant systems.

Minor comments:

The authors refer to a "green" photosynthesis system. I recommend against using the term "green" in this context because it is too vague for a scientific paper.

The requirement of this system for NADH can be a drawback because this is an expensive reagent, but if it is recycled efficiently, this concern is addressed. The authors should assess how much the NAD⁺/NADH is turned over in the system and what the yield of H₂ is for NADH added.

Activity is said to be made possible by placing the reactive groups in close proximity within a microenvironment. But I do not see a comparison to an analogous system with freely diffusing components – such a comparison would help back up this claim.

For Reviewer #1

Comments: The manuscript entitled “A host–guest semibiological photosynthesis system: coupling artificial and natural enzymes for solar alcohol splitting” describes a novel combination of synthetic and enzymatic systems for photocatalytic hydrogen evolution via alcohol dehydrogenation. The authors appear to build on the success of their recent efforts (Nat. Commun. 2020, 11, 2903–2913) to enhance photocatalysis through biohybrid constructs in which a metallo-organic artificial enzyme can be regenerated by a biological enzyme to which it is bound. While this investigation offers interesting and promising insights on how to design supramolecular biohybrid assemblies for coupled catalysis, there are a number of major and minor issues with the manuscript that may inform the authors’ and editors’ revisions and decisions.

MAJOR ISSUES:

Reviewer’s Comments (1): Throughout this work (and the previous Nature Communications paper by the same team), the authors use the terms “host-guest” and “matryoshka” to describe the binding of the metallo-organic cage “inside” the enzyme’s catalytic domain. This internal binding is a central claim of the paper. In order to use these terms, the authors should prove that cage binds “inside” a protein cavity rather than “outside” on an external surface of the protein. What experimental evidence is there for this “inside” binding of the cage to the enzyme? I can only find support in the form of the authors’ claim that “Co–TPS (10.0 μ M) at 300 nm decreased linearly when treated with ADH suggesting that the artificial catalyst was shielded by the cavity of ADH⁵⁵”. However, I suspect this linear decrease (Figure S6) is simply attributable to dilution. If not, the authors need to substantiate the claim, because their citation of reference 55 does not explain or support it. Reference 55 does not discuss how a linear decrease in UV absorption supports endohedral binding (indeed, quite the opposite, the authors of that paper admit they cannot obtain binding data from their UV-Vis spectra on page S27 of their supplemental document) nor does the paper involve ADH or any protein binding whatsoever. It is not unlikely that the Co–TPS cage binds to the ADH protein via multiple non-specific surface interactions involving hydrogen bonding and charge-pairing interactions, which are likely strengthened by the presence of a large proportion of organic solvent in their experiments. This counter-hypothesis to the “inside” binding claim does not necessarily preclude the artificial enzyme from enhanced catalytic function by virtue its proxy to the enzyme’s active site, and could also explain why Co–TPS interacts with the protein more strongly than Co–BDT (if that is the case – the Co–BDT-protein interaction is not discussed in the paper), since Co–BDT has far

fewer charged and hydrogen bonding sites than Co-TPS.

Responses: Thanks very much for the thoughtful suggestions and sorry for the ambiguous citation. Actually, our target is to anchor the artificial catalyst spatially close to the active sites of the natural enzyme, then combine NADH-mediated photocatalytic hydrogen evolution and NAD⁺-mediated enzymatic alcohol dehydrogenation *in situ*. As reviewer considered, our manuscript initially focused on the encapsulation of artificial metal-organic cage with its guest into the enzyme catalytic pocket. Indeed, our experiments results have provided positive information to support our assumption:

(1) Isothermal titration calorimetry (ITC) of the enzyme ADH upon the addition of cage Co₃TPS₂ (labeled as Co-TPS in the original manuscript) exhibited a 1:1 enzyme/cage complexation species with a disassociation constant of 0.23 μM (Fig. 5a); Luminescence titration of dye-encapsulated cage [Co₃TPS₂ ⊃ PNQ]²⁻ upon the addition of ADH exhibited significant quenching of luminescent intensity, and the titration curve was coinciding with the Hill plot (Fig. 5d).

(2) Theoretical docking results suggested that the cage Co₃TPS₂ was capable of perching on the enzymatic pocket of ADH and burying its active moiety into ADH to form a working module (Supplementary Fig. 33). Random samples of the combination of Co₃TPS₂ and ADH showed that the cage perched on enzymatic pocket is the major conformation of the host-guest species (Supplementary Fig. 34).

(3) The pseudo-zero-order kinetic behavior of the combined system during the initial stage of reactions inferred that the substrate and the cage were all included into the pocket of the enzyme ADH (Fig. 4). The appended kinetics experiments inspected that the initial rates of the reaction satisfied a Lineweaver-Burk plot with concentrations of NAD⁺ or Co₃TPS₂ (Supplementary Fig. 27). The typical Michaelis-Menten kinetics reflected that enzyme/cage complexation species should be considered as a new enzyme based on both the structure and the catalytic function.

The citation of reference 55 in the original manuscript provided a possibility that the decrease in UV-Vis absorption spectra might be owing to the encapsulation and shielding effect of ADH on Co₃TPS₂. As for the UV-Vis absorption titration, we repeated the titration experiments. A blank experiment that a total 20 μL of blank solvent added to a 3 mL cell did not cause obvious spectra changes, demonstrating that the dilution effects were negligible. The UV-Vis linear change in the binding process also disclosed in the updated reference 55 (reference 56 in the original manuscript). Appended UV-Vis difference spectra of the cage Co₃TPS₂ upon the addition of ADH revealed a significant spectra changes centered at 268 nm from an initial

linear growth to almost unchanged (Supplementary Fig. 7). In this case, we thought that ADH and the cage Co_3TPS_2 possibly interacted to form an enzyme/cage complexation species.

We also performed additional experiments to monitor the dehydrogenation of alcohol to form aldehyde. Under identical conditions containing ADH ($10 \text{ U}\cdot\text{mL}^{-1}$) and NAD^+ (2.0 mM), the presence of the mononuclear reference compound CoBDT_2 (labeled as Co-BDT in the original manuscript) led to a significant inhibition of the dehydrogenation, but the cage Co_3TPS_2 hardly affected the formation of aldehyde (Supplementary Fig. 28). Alternatively, under light irradiation, the addition of **PNQ** dye to the aforementioned systems enhanced the efficiency of the enzyme/cage complexation systems, but did not positively affect the $\text{CoBDT}_2/\text{ADH}$ catalytic systems (Fig. 3b). The simultaneous binding of NADH and the dye **PNQ** in the cage cavity gave a 1:1:1 ternary complex, and this 1:1:1 assembly interacted with the dehydrogenase ADH active site to form such a suprastructure, which was essential for the efficient coupling of the two halves of the tandem catalysis. We added some words corresponding to the appended experiments in the revised manuscript to describe our results more clearly.

Reviewer's Comments (2): The authors do offer a computational evidence for the “inside” docking in the form of a theoretical calculation, but this calculation is not compared energetically with non-specific “outside” docking interactions, and even these models (Fig. S29) appear to show the cage at the proteins surface, with only the NADH/NAD^+ molecules in the catalytic domain.

Responses: Thanks very much for the thoughtful suggestions. As mentioned by reviewer, ADH bound Co_3TPS_2 through non-covalent interactions, which was conducive to simplifying the assembly processes through supramolecular interactions, despite of that Co_3TPS_2 might combine in different positions of ADH. In the revised manuscript, we used new models in which Co_3TPS_2 randomly combined with ADH for theoretical docking study (Supplementary Fig. 34). The results showed that the cage located at the enzyme catalytic pocket was the major binding state with a binding energy over $40.0 \text{ kJ}\cdot\text{mol}^{-1}$. Only a few samples showed that Co_3TPS_2 bonded outside the catalytic pocket of ADH with a lower binding energy. We added several words in the revised manuscript to describe these results that Co_3TPS_2 was probably perched on enzymatic pocket of ADH to gain a more stable binding state due to the contribution of non-covalent interactions.

Reviewer's Comments (3): Figure 1: The cartoon of the “artificial enzyme” in Fig. 1 and Fig. 6 is confusing; it doesn't show a representation of the prismatic Co-TPS cage; only the NADH molecules the cage allegedly “traps” – this drawing initially misleads the reader into thinking

that the NADH molecules are the ligands of the cage bound to cobalt, which is not true. The authors should revise their schematics to more accurately reflect the structure of the artificial enzyme complex.

Response: Thanks very much for the careful suggestions. According to the reviewer's suggestions, we have carefully modified the Fig. 1 and Fig. 6. We have replenished the assembly procedure of Co_3TPS_2 in Fig. 1, and showed a representation of the prismatic cage Co_3TPS_2 . In addition, the ambiguous contents were well modified. Likewise, we have added remarks for substances appeared in Fig. 6, and enriched the description of the enzymatic processes.

Reviewer's Comments (4): If NADH can be added to PNQ@Co-TPS to form a ternary assembly, could it not also add a second and third time to form quaternary and quinary assemblies too? This issue seems not to be discussed even while the cartoon in Fig. 1 seems to imply it. The ITC data seems to be fit to a 1:1 binding of NADH to $\text{PNQ@Co}_3\text{TPS}_2$; why doesn't NADH bind a second and third time to the other bay regions of the host? The citations 23, 49, and 50 on line 257 do not offer an obvious explanation.

Response: Thanks very much for the thoughtful suggestions. As mentioned by the reviewer, we did observe the combination of multiple guest molecules in another work of ours (reference 32). However, in this study, ITC data provided the result of a 1:1 binding of NADH to $\text{PNQ@Co}_3\text{TPS}_2$ (Fig 5a), which could be further verified by Benesi-Hildebrand fitting of the UV-Vis spectra (Supplementary Fig. 5). Of course, we cannot rule out the possibility of further forming quaternary and quinary assemblies, but these assemblies might be unstable and did not significantly impact on the catalysis.

Reviewer's Comments (5): Several of the supplementary figures are not mentioned or discussed at all in the main text nor the supplementary information file. These figures should be cited and discussed or else removed entirely if they are not worth talking about. (Figures S4 S5 S9 S13 S14, possible others).

Response: Many thanks to the reviewer for the suggestions. We have cited and discussed all figures appearing in the revised supplementary file at the corresponding positions in the new version of manuscript.

Reviewer's Comments (6): Lines 188-190: "This result suggested that the proton reduction catalysis driven by Co-BDT interfered with the enzymatic reactions, since a stable supramolecular host-guest semibiological system could not be realized." How do the authors know a stable host-guest system was not realized for Co-BDT ? There is no supporting data (e.g., ITC) to show that Co-BDT does not bind ADH. Furthermore, is it possible that binding "inside"

by Co-BDT could be responsible for this inhibitory effect by blocking reactive site access for substrates?

Response: Thanks very much for the thoughtful suggestions. First, we performed the luminescent titration of NADH to the solution containing both CoBDT₂ and PNQ, the intermolecular quenching with a Stern-Volmer constant calculated as 6650 M⁻¹ suggested a typical homogeneous behavior that was different from that of the Co₃TPS₂/PNQ/NADH supramolecular system (Supplementary Fig. 9). ITC experiment of ADH upon the addition of CoBDT₂ gave a disassociation constant of 13.3 μM (Supplementary Fig. 18), which agreed with the assumption of the reviewers. Dehydrogenation experiment of alcohol to form aldehyde utilizing ADH (10 U·mL⁻¹) and NAD⁺ (2.0 mM) exhibited that the mononuclear reference compound CoBDT₂ could significantly inhibit the reaction, which might be due to the inert binding between ADH and CoBDT₂ (Fig 3b; Supplementary Fig. 28). We thus added some words to describe the aforementioned experimental results and modified related statements in revised manuscript to illustrate the main roles of the mononuclear compounds. In this case, the formation of enzyme/cage/dye complexation species would efficiently couple the two halves of the tandem catalysis, whereas the reference mononuclear compound inhibited the active sites of enzyme and hardly combined the two halves' conversions.

Reviewer's Comments (7): Why are there no error bars in any of the quantitative data (Figs. 3b-c, 4, 5a, S14-S25)? Does it mean the experiments each performed only once? If so, why should readers trust the data from one-time experiments that were not repeated?

Response: Thanks very much for the careful suggestions. Before submitting the manuscript, we have checked and repeated the dates with respect to catalysis and ensured the quantitative data credible. According to the reviewer's comments, we reconfirmed the experiments related to quantitative data and added error bars in related figures for the accuracy of the statements.

MINOR ISSUES:

Reviewer's Comments (1): The authors start several sentences with "And" (lines 15, 41, 112, 279, 326) which is awkward and best avoided.

Response: Thanks very much for the careful suggestions. We have deleted the word "And" in the relevant sentences to improve the quality of writing.

Reviewer's Comments (2): Line 98: Why do the authors cite reference 41 when they claim their cage is appropriately shaped for binding PNQ via aromatic stacking? Reference 41 does not discuss PNQ binding, while there are many prior works by Makoto Fujita describing aromatic stacking in metallacages more generically. Reference 42 is also questionable in this regard as it

discusses DNA intercalation.

Response: Thanks very much for the careful suggestions and sorry for the improper references. Although the references 41 and 42 in the original manuscript might show that aromatic stacking was beneficial to the formation of host-guest species, these citations are not accurate enough as pointed out by reviewer. To make this point more clearly, we have added some word in the revised manuscript and replaced the references 41 and 42 in the original manuscript with other reasonable citations (reference 35 and 36 in the revised manuscript) for suggesting that the guest molecule **PNQ** possessing aromatic planes was capable of forming a sandwich host-guest complex with the pillared host Co_3TPS_2 in a face-to-face way.

The supplementary references in the revised manuscript were shown below:

35. Kumazawa, K., Biradha, K., Kusukawa, T., Okano, T. & Fujita, M. Multicomponent assembly of a pyrazine-pillared coordination cage that selectively binds planar guests by intercalation. *Angew. Chem. Int. Ed.* **42**, 3909–3913 (2003).
36. Nakamura, T., Ube, H. & Shionoya, M. Silver-mediated formation of a cofacial porphyrin dimer with the ability to intercalate aromatic molecules. *Angew. Chem. Int. Ed.* **52**, 12096–12100 (2013).

Reviewer's Comments (3): Page 4: Should the authors refer to the CoS_4 cores as CoS_4^- cores to indicate the cobalt (III) ion and not imply a Cobalt (IV) oxidation state?

I recommend the authors refer to the cage as Co_3TPS_2 or $\text{Co}_3(\text{TPS})_2^{3-}$ instead of Co-TPS , since the latter may mislead the reader into thinking there is an equal portion of Co ions and **TPS** ligands.

Reference 39, change “Common.” to “Commun.” (Line 610)

Reference 43, change “Nelsonbc” to “Nelson”.

Response: Thanks very much for the careful and thoughtful suggestions. We have changed “ CoS_4 cores” to “cobalt dithiolene core/moiety” in the revised manuscript for stating more accurately. In addition, we were willing to adopt the suggestions of reviewers and change “ Co-TPS ” to “ Co_3TPS_2 ”. In the meanwhile, the “ Co-BDT ” was changed to “ CoBDT_2 ” for clarifying the number of metals and ligands within the catalyst. We are so sorry for the spelling mistakes and we have corrected these typos in the revised manuscript and carefully checked full manuscript for improving the quality of manuscript.

Reviewer's Comments (4): The synthetic details in the SI are confusing. For example:

- In the synthesis of H_6TPS , compound **5** is referred to by its IUPAC name on page 6, line 100, even though the IUPAC name is not mentioned earlier in the synthesis of compound **5**. The

authors should stick with “compound 5” or define compound 5 by the IUPAC name the first time it is used.

Response: Thanks very much for the careful suggestions. We are sorry for our irregular writing. We made modifications in the revised SI based on suggestions and stuck with “compound 5” to define the corresponding compound, and all changes were highlighted in yellow.

Reviewer’s Comments (5): SI Page 6, Line 113, what is “precursor”? Presumably it is H₆TPS, but the authors should be specific about which reagents they are using in a procedure.

Response: Thanks very much for the careful suggestions. The description of “precursor” mainly refers to the reference 59. To obtain the ligand H₆TPS requires the thioethers deprotection reaction of the precursor. To make the statement more clearly, we have revised the Supplementary Fig. 1 (Scheme S1) and specified the reagents used based on reviewer’s suggestions.

Reviewer’s Comments (6): Line 103, the “H₆TPS precursor” is described as a “pure product” but then it is purified to afford “H₆TPS ligand” in lines 112-123. Why do the authors call it a pure product if this purification step is required prior to adding cobalt?

Response: Thanks very much for the careful suggestions. The H₆TPS precursor is a pure product as shown in revised Supplementary Fig. 1. This precursor needs to undergo a deprotection step (rather than purification step) to obtain the ligand H₆TPS. As the ligand H₆TPS was extremely unstable, so it was used directly without further purification for the next coordination reaction.

Reviewer’s Comments (7): Please specify which reagents are variable in the actual graphs of Fig. 4 so readers do not have to search the captions to find the information.

Response: Thanks very much for the careful suggestions. We have specified the reagents that are variable in the actual graphs of Fig. 4 in the revised manuscript.

Reviewer’s Comments (8): Figure 5a is difficult to read because it has 3 axes. Perhaps the plots can be fit on a single axis with insets to zoom in on the lower-amplitude curves.

Response: Thanks very much for the kind remind. To elaborate more clearly, we have made revisions based on the reviewer’s suggestions.

For Reviewer #2

Comments: This is a well-done and interesting example of a supramolecular system which combines biotic and artificial components to effect simultaneously light-induced oxidation and reduction processes. The reduction (at the artificial Co/dithiolene redox site) is of two protons to H₂; the coupled oxidation is of ethanol to the aldehyde by an alcohol dehydrogenase enzyme. Thus, the overall result is light-induced conversion of ethanol to aldehyde + H₂, conceptually similar to photosynthetic water-splitting.

The coupling of the two redox cycles has been the focus of much recent work from this (and other groups) in recent years. The work reported here constitutes a very nice example which is a significant advance in this field and it deserves publication in Nature Comms. The work is meticulously thorough and the conclusions well-evidenced with all necessary control experiments in place; in particular the simultaneous binding of NADH and the sensitiser in the host cavity to give a 1:1:1 ternary complex, an essential prerequisite for this process to work, has been established by spectrophotometric and ITC titrations. Further, there is evidence that this 1:1:1 assembly interacts with the dehydrogenase enzyme active site which is also essential for efficient coupling of the two halves of the process.

I have just one question for the authors to consider. The reaction progress is monitored by formation of H₂, which is relatively easy to measure. Of course, this should be accompanied by formation of an equivalent amount of aldehyde. If there is some way to demonstrate aldehyde formation, and confirm that the quantity matches the H₂ production, that would be very nice. There are various possible methods (maybe IR, GC-MS, NMR...) but simplest might be a colorimetric assay. Subject to consideration of this possibility I therefore recommend acceptance of this interesting and well-done study.

Response: Thanks very much for the thoughtful and kind suggestions. According to the reviewer's suggestions and references 17/66, we have added some additional experiments to monitor the formation of aldehyde through the gas chromatography using external standard method. As shown in Fig. 3b, the photocatalysis in EtOH/H₂O (3:2) solution containing **PNQ** (0.5 mM), Co₃**TPS**₂ (labeled as Co-**TPS** in the original manuscript, 20.0 μM), ADH (10 U·mL⁻¹) and NAD⁺ (2.0 mM) produced hydrogen and aldehyde simultaneously. The similar amount of the two products showed a fine synergic combination of abiotic and biotic synthetic sequences for photocatalytic fuel and chemical transformation. Notably, the production of aldehyde was always higher than hydrogen production, suggesting that the coenzyme NADH played an important role in storing protons and electrons (Supplementary Fig. 29). This stable electron and hydride transfer process eliminated the demand for transferring protons and electrons generated during

the reaction immediately, endowing the catalytic cycle with redundancy reminiscent of natural photosynthetic system. Related description was added in corresponding place of the revised manuscript and highlighted in yellow.

The supplementary reference in the revised manuscript were shown below:

66. Simon, T., Bouchonville, N., Berr, M. J., Vaneski, A., Adrović, A., Volbers, D., Wyrwich, R., Döblinger, M., Susha, A. S., Rogach, A. L., Jäckel, F., Stolarczyk, J. K. & Feldmann, J. Redox shuttle mechanism enhances photocatalytic H₂ generation on Ni-decorated CdS nanorods. *Nat. Mater.* **13**, 1013–1018 (2014).

For Reviewer #3

Comments: The authors describe a host-guest system that couples the action of the enzyme alcohol dehydrogenase with a dithiolene-embedded MOF hosting an organic dye as a photosensitizer. This system the yields H₂ through NADH-mediated hydrogen production, where NADH (and a proton) result from ethanol dehydrogenation (alcohol + NAD⁺ → aldehyde + NADH + H⁺). Hydrogen production is catalyzed by an “artificial enzyme” consisting of a cobalt dithiolene complex embedded in a MOF, which also contains the ADH enzyme and the photosensitizing dye. Overall, I like this approach that combines MOF chemistry, Biochemistry, coordination chemistry, and photocatalysis for solar fuels. I find the approach to be quite clever.

Reviewer’s Comments (1): It took me many reads of the manuscript to understand the system in place here because the presentation of the system is confusing. One major issue around clarity is that the structure and composition of the cobalt catalyst is not made clear early in the paper. While I came to understand through reading further that the cobalt dithiolene sites in the MOF, described as Co-TPS, are catalytic sites for hydrogen evolution, it was difficult to figure out whether Co-TPS is a MOF node or a catalyst or both. It is not clear to me what the structure is that is shown as the “artificial enzyme” in Fig. 1? Is that a different cobalt catalyst? Also, what TPS stands for was not defined. Fig. 1 overall is confusing. Does the alcohol dehydrogenase (red outline) encapsulate the PNQ dye and also the cobalt active site for H₂ production? What is the relationship between the MOF structure and the enzyme?

Response: So sorry for the ambiguity in the manuscript and many thanks for the careful reminds and suggestions. In order to illustrate the structure and composition of the cobalt catalyst and its binding relationship with the natural enzyme ADH more clearly, we have carefully modified the Fig. 1 in the manuscript and given a more detailed description of the relevant content. Actually, the cage Co₃TPS₂ (labeled as Co-TPS in the original manuscript, where TPS stands for the deprotonated ligand H₆TPS) is an individual molecular triangular prism that could dissolve in the solution, rather than a metal-organic framework which is a crystalline solid. The cobalt dithiolene fragments in the cage Co₃TPS₂ are both the connecting nodes and electrochemical catalytic sites, and the cage Co₃TPS₂ consists of a cavity reminiscent to the pocket of natural enzyme. The active catalytic sites and the individual hydrophobic cavity of Co₃TPS₂ makes this kind of artificial host be portrayed as “artificial enzyme”. The cage Co₃TPS₂ are capable of encapsulating the positively charged dye PNQ within its hydrophobic cavity. And this complexation species was further included into the pocket of enzyme ADH to form a matryoshka type supramolecular catalyst, which combines the hydrogen evolution and dehydrogenation of alcohol to form aldehyde in situ under light irradiation.

Reviewer's Comments (2): There are some statements in the manuscript that are not well supported. The CoS₄ sites in the MOF are said to yield “outstanding redox activity with a particularly low overpotential.” How is outstanding redox activity defined and measured? Electron transfer rates or some other measure? I do not see any overpotential measurement either or indication of how the overpotential here is referenced (with respect to proton reduction...under what conditions and using what reagents?) The authors report a CV of the Co-TPS (Fig. 3a) and relate it to cobalt dithiolene species but that is a reversible CV, not a catalytic CV, and no substrate is added to test whether this redox couple supports catalysis (assuming this is collected in aprotic solvent, as suggested by the electrolyte given, but solvent is not indicated here – if this is collected in H₂O/EtOH, then this redox couple is not likely involved in catalysis).

Response: Thanks very much for the kind reminds and careful suggestions. We are sorry for the incomplete citation here. As mentioned in the literatures (references 33/34/41 in the revised manuscript), the cobalt dithiolene species (CoS₄) always possessed outstanding redox activity. In the reference 34, the overpotentials of 0.34 and 0.53 V are required for MOS 1 and 2, respectively, to reach current densities of 10 mA/cm² at pH 1.3 with respect to proton reduction in fully aqueous conditions. Under the electrochemical condition, Co₃TPS₂ exhibited an overpotential of 0.16 V for proton reduction (Supplementary Fig. 12, determined by the method of Evans referring to references 42/43). The electrochemical experiments of both Co₃TPS₂ and CoBDT₂ (labeled as Co-BDT in the original manuscript) in aprotic solvent DMF revealed the reversible redox behavior consistent with reference 33 (Fig 3a), wherein, the addition of trifluoroacetic acid triggered the appearance of a catalytic wave and the cathodic peak current had a linear dependence on catalyst concentration (Supplementary Figs. 11–14). These results all supports our postulation, and we added some words and references in the revised manuscript, and modified the related descriptions.

The supplementary references in the revised manuscript were shown below:

34. Clough, A. J., Yoo, J. W., Mecklenburg, M. H. & Marinescu, S. C. Two-dimensional metal organic surfaces for efficient hydrogen evolution from water. *J. Am. Chem. Soc.* **137**, 118–121 (2015).
41. McNamara, W. R., Han, Z., Yin, C., Brennessel, W. W., Holland, P. L. & Eisenberg, R. Cobalt-dithiolene complexes for the photocatalytic and electrocatalytic reduction of protons in aqueous solutions. *Proc. Natl. Acad. Sci. USA* **109**, 15594–15599 (2012).
42. Felton, G. A. N., Glass, R. S., Lichtenberger, D. L. & Evans, D. H. Iron-only hydrogenase mimics. Thermodynamic aspects of the use of electrochemistry to evaluate catalytic

efficiency for hydrogen generation. *Inorg. Chem.* **45**, 9181–9184 (2006).

43. Helm, M. L., Stewart, M. P., Bullock, R. M., DuBois, M. R. & DuBois, D. L. A synthetic nickel electrocatalyst with a turnover frequency above 100,000 s⁻¹ for H₂ production. *Science* **333**, 863–866 (2011).

Reviewer's Comments (3): It is clear that this system works, but the activity is not too impressive, but also not made very clear. In Fig. 4, the authors the volume of H₂ produced over time and my quick comparison to other systems in the literature is that it on the order of what is typically seen in systems combining Ru(bpy)₃²⁺ with a molecular catalyst and ascorbic acid. However, the authors do not compare their activity to other systems, and do not give a turnover number or information such as initial rate. They give some information on initial rate in Figure S23-25 as a function of conditions but again these results are not compared to other systems in the literature. Also, critically, I do not find a turnover number reported anywhere in the manuscript or in the SI. Without a turnover number, it is not even verified that the system is truly catalytic (although I suspect it is based on the data shown, nevertheless a TON is needed). A TON for production of H₂ with respect to catalyst and photosensitizer both could be determined, assuming that each cobalt site available can be a catalytic site; TON with respect to ADH also could be reported. I realize that TON is tricky to define in such a system but it can't be ignored.

Response: Thanks very much for the thoughtful suggestions. In fact, to the best of our knowledge, the Co₃TPS₂/PNQ/NADH supramolecular system is one of the few homogeneous photocatalytic systems that can use NADH as a direct electron donor to achieve artificial proton reduction, and the photocatalysis driven by non-noble homogeneous catalyst Co₃TPS₂ exhibited similar catalytic conversion to that of noble metal catalytic system with 18% consumption of electron donors.

We provided H₂-evolution TON values 45 and 1125 with respect to cobalt in Co₃TPS₂ and ADH to clarify the catalytic status of the system. Based on the TON values of the hydrogen evolution, our supramolecular system exhibited the comparable efficiency to those previous reported similar supramolecular catalytic systems (references 32/46/47). The comparison and the new experimental results were added in the revised manuscript.

The supplementary references in the revised manuscript were shown below:

46. He, C., Wang, J., Zhao, L., Liu, T., Zhang, J. & Duan, C. A photoactive basket-like metal-organic tetragon worked as an enzymatic molecular flask for light driven H₂ production. *Chem. Commun.* **49**, 627–629 (2013).
47. Yang, L., Jing, X., He, C., Chang, Z. & Duan, C. Redox-active M₈L₆ cubic hosts with

tetraphenylethylene faces encapsulate organic dyes for light-driven H₂ production. *Chem. Eur. J.* **22**, 18107–18114 (2016).

Reviewer's Comments (4): The proposed mechanism is quite rudimentary. That's not a problem as Fig. 6 is a good clarifying figure, but the work gives minimal real insight into mechanism. For example, the key fundamental question of whether photosensitizer quenching is reductive or oxidative is not even addressed. I also am concerned about the authors proposing Co(II) to be the active species in proton reduction (Table 1), which is not supported.

Response: Thanks very much for the thoughtful suggestions. Briefly, the total conversion containing two half reactions, one is the light-driven hydrogen evolution, the other is the dehydrogenation of alcohol. The mechanism of the former has been reported in the references 28/45 shown below, whereas, the latter is a well-known enzymatic process. We utilized the recovery conversion of NAD⁺/NADH couple to connect the two half reactions synergistically.

As for the question that the photosensitizer quenching is reductive or oxidative, and what the active species in proton reduction is (Table 1), the detailed investigations have been reported in reference 28/45 and reference 33/44, respectively. Inspired by the discussion in the literatures, we thought that the closed proximity between the photosensitizer **PNQ** and the electron donor NADH by host-guest approaches would be conducive to accelerating the photoinduced electron transfer from NADH moieties to the excited photosensitizer, giving a long-lived reduced photosensitizer to further reduce the cobalt ions on Co₃TPS₂ for proton reduction and hydrogen evolution. Based on the electrochemical results, it could be concluded that Co₃TPS₂ was capable of reducing protons through an ECCE pathway (E corresponds to an electron transfer step and C to a protonation reaction), in which doubly protonated Co(II) dithiolene species undergoes the process of proton transfer from a protonated sulfur to the cobalt core to form active Co(III)-H species after receiving the second electron. We have revised the related statements and Table 1 to provide a reasonable catalytic mechanism, and supplied the well-supported references in corresponding place of the revised manuscript.

The supplementary references in the revised manuscript were shown below:

44. Queyriaux, N. Redox-active ligands in electroassisted catalytic H⁺ and CO₂ reductions: benefits and risks. *ACS Catal.* **11**, 4024–4035 (2021).
45. Fukuzumi, S., Hong, D. & Yamada, Y. Bioinspired photocatalytic water reduction and oxidation with earth-abundant metal catalysts. *J. Phys. Chem. Lett.* **4**, 3458–3467 (2013).

In summary, this manuscript is much too preliminary to be published. It presents a potentially interesting system but its activity is not well characterized and the approach and analysis is not

clearly explained to the reader. The investigation of the mechanism neglects fundamental questions of how electron transfer occurs in the system. The electrochemical study looks only at a reversible couple likely irrelevant to catalysis. Finally, the activity is not placed into context of the field through comparison to other relevant systems.

Minor comments:

Reviewer's Comments (1): The authors refer to a “green” photosynthesis system. I recommend against using the term “green” in this context because it is too vague for a scientific paper.

Response: Thanks very much for the careful suggestions. The term “green” in this context originally refers to the use of low-toxic solvents under catalytic conditions and the circumvention of high temperature and pressure conditions required in alcohol reforming processes. Since the term “green” in this context is vague, we adopted the reviewer's suggestions and revised the relevant statement to make the expression clear.

Reviewer's Comments (2): The requirement of this system for NADH can be a drawback because this is an expensive reagent, but if it is recycled efficiently, this concern is addressed. The authors should assess how much the NAD^+/NADH is turned over in the system and what the yield of H_2 is for NADH added.

Response: Thanks very much for the thoughtful suggestions. Indeed, it is our ideal to pursue NADH's continuous recycling in the catalytic system. According to the reviewer's suggestions, we assessed the conversion of NAD^+/NADH based on the production of aldehyde (19.2 μmol) and total 192% NAD^+ was turned over in the system, and the production of H_2 (296 μL) suggested the yield of H_2 was up to 132% (based on the coenzyme couple concentration). We anticipated that this conversion could continue after prolonging the reaction time with additional catalysts. We have revised the related statements in corresponding place of the revised manuscript.

Reviewer's Comments (3): Activity is said to be made possible by placing the reactive groups in close proximity within a microenvironment. But I do not see a comparison to an analogous system with freely diffusing components – such a comparison would help back up this claim.

Response: Thanks very much for the thoughtful suggestion. Indeed, we employed mononuclear compound CoBDT_2 resembling a corner of the Co_3TPS_2 catalyst as a freely diffusing component to suggest that the microenvironment provided by Co_3TPS_2 contributed to the activity of the abiotic–biotic hybrid systems. The use of CoBDT_2 (60.0 μM , ensuring the same concentration of cobalt ions) yielded only 7 μL hydrogen following a shorter life time of 12 h under same reaction conditions (Fig. 3b and Table 1, entry 7), despite the fact that the redox potential (-0.56 V vs.

Ag/AgCl) of CoBDT₂ was identical to that of Co₃TPS₂ (Fig. 3a). Moreover, an inhibition experiment was further carried out by adding a non-reactive species, 1,1-dimethyl-1,2,3,4-tetrahydro-quinolinium salts (DTQ) that is similar in configuration to the photosensitizer PNQ could compete to occupy the cavity of Co₃TPS₂. The addition of inhibitor DTQ (0.1 M) into the optimal reaction system resulted in an effective quenching of the catalysis with a hydrogen yield that was only 17% of the original system (Fig. 3c and Table 1, entry 8). In addition, the initial rate of aldehyde production driven by Co₃TPS₂-based semibiological system was estimate to 1.14 μmol·h⁻¹, a rate was higher than that of blank system without abiotic module (0.89 μmol·h⁻¹). These results indicated that Co₃TPS₂ created an isolated artificial catalytic microenvironment, and the formation of localized catalysis allowed in-situ communication between the abiotic and biotic systems through coenzymes, which was superior to the free diffusion system.

REVIEWER COMMENTS

Reviewer #1 (Remarks to the Author):

Many of my concerns with the initial manuscript regarded the claims and data around the various host-guest binding events. In my second review of the manuscript, I cannot find a description of the equations and fitting methods that were used to estimate stoichiometries and binding constants for any of the data except for the ITC data. For example, Supplementary Fig. 7 appears to show a non-linear fit of UV-Vis absorption difference spectra, whereas most of the other fitting methods (besides ITC) appear to be based on linear fits. The authors should be informed that linear regression methods such as Benesi-Hildebrand fitting (Supplementary Fig. 5) and Stern-Volmer (Supplementary Figs. 8-9) are outdated, as better non-linear regression methods are now available. See the highly cited review by Thordarson (Chem. Soc. Rev. 2011, 1305). In particular the authors might consider acknowledging the possibility of other binding stoichiometries for the binding of NADH beyond 1:1 without robust evidence to the contrary. In general, however, the authors have adequately addressed my concerns from the initial review in their revised manuscript and rebuttal letter. I do not have major concerns about publishing this manuscript.

Reviewer #2 (Remarks to the Author):

The authors have done a careful and thorough job of revising the paper in light of the comments from the first three reviewers. These revisions include

- more detailed electrochemical studies
- discussion of the possibility of the cage/guest system binding on the protein external surface, including additional computational docking studies and UV/Vis spectroscopic analysis
- explicit analysis of aldehyde formation by GC
- indication of errors in experimental data
- calculation and inclusion of turnover numbers for catalysis
- confirmation of the supramolecular catalytic nature of the process using an inhibitor which blocks the cage cavity and removes most catalytic activity
- numerous small corrections to text, figures and references to clarify minor issues or lack of clarity in some cases.

The resulting extensive revision is significantly improved and I am happy to support publication. The conclusions are well evidenced and the study will attract significant interest.

Mike Ward

Reviewer #3 (Remarks to the Author):

The authors have revised their manuscript to address a number of points raised. While the manuscript is improved, are many outstanding questions about the characterization of the system and reactivity that cause the manuscript still be speculative. Furthermore, the enzyme portion of the system is not well described and not characterized.

To better understand the Co3TPS2 unit's interaction with ADH, the authors have performed a docking study and how describe the Co3TPS2 cage as "perched on enzymatic pocket." From a biochemical standpoint, this description of perched on a pocket does not make sense. Enzymatic pockets are buried – it sounds like, based on the docking study, part of the complex is buried and part contacting the surface near the pocket(?) It is recommended that the authors consult with biochemists or

structural biologists regarding the characterization and description of this complex to use the best terms to describe this proposed interaction. The authors provide Fig. 1 with a structure of a host-guest semibiological system with the metal-organic cage placed in the ADH active site. However, there is no information in the figure caption on how this structure shown is made. Presumably, this is from the docking study, but this needs to be specified. Looking at the docking study results in the SI, they seem to show the cage associated with the enzyme surface from what I can see but it is hard to discern from these results where it is located relative to the active site as no landmarks are given. Critically, I can find no information on the origin of this ADH structure (what is the PDB code?) or assumptions made in the docking study. Regarding the experimental studies of ADH, I can't find information about how the ADH samples were obtained and characterized in this study, or, critically, even what their source is (human? yeast? what isozyme?)

To the best of my knowledge, ADH is a dimer (sometimes a tetramer) and the dimer is thought to be the functional unit. However, the relationship of the Co3TPS2 to the full ADH structure as a dimer with its two Zn sites is not described. Furthermore, there is no characterization of this structure that reveals the state of the ADH and the ADH-Co3TPS2 complex. A number of straightforward experiments can and should be done -- UV-vis and CD of the enzyme along compared to the enzyme with Co3TPS2, hydrodynamic radius measurement (i.e. gel filtration) of enzyme and enzyme + Co3TPS2, and dynamic light scattering, at least. Otherwise, the structure presented through the docking study is a proposed but not verified structure. I regret that I did not raise these issues previously, but I found the original manuscript too confusing and had difficulty figuring out what the system was that was being studied in order to make specific suggestions around characterization.

The authors have now presented more information on the H₂ evolution activity of the Co3TPS2 complex, and draw conclusions on the mechanism, calling it an ECCE pathway. There is no experimental data that can support that mechanism in this paper. Indeed, distinguishing between ECEC and ECCE is usually not possible even in dedicated, detailed electrocatalysis studies of H₂ production. Further speculation that they present regarding protonation of the dithiolene species and formation of a Co(III)-H is also not supported.

In summary, the authors present a creative approach to photocatalytic H₂ production. However, there are standard, fundamental characterization steps missing in this work. Description of the ADH samples and details of the docking study setup and results are missing. Furthermore, the conclusions continue to be much too speculative (in particular regarding mechanism). As a result, the paper continues to include greatly overinterpreted data and speculation. If the stuck to presenting their results as they are and drawing conclusions that can be well supported by the data, this would be an interesting paper worth publishing.

For Reviewer #1

Reviewer's Comments: Many of my concerns with the initial manuscript regarded the claims and data around the various host-guest binding events. In my second review of the manuscript, I cannot find a description of the equations and fitting methods that were used to estimate stoichiometries and binding constants for any of the data except for the ITC data. For example, Supplementary Fig. 7 appears to show a non-linear fit of UV-Vis absorption difference spectra, whereas most of the other fitting methods (besides ITC) appear to be based on linear fits. The authors should be informed that linear regression methods such as Benesi-Hildebrand fitting (Supplementary Fig. 5) and Stern-Volmer (Supplementary Figs. 8-9) are outdated, as better non-linear regression methods are now available. See the highly cited review by Thordarson (Chem. Soc. Rev. 2011, 1305). In particular the authors might consider acknowledging the possibility of other binding stoichiometries for the binding of NADH beyond 1:1 without robust evidence to the contrary. In general, however, the authors have adequately addressed my concerns from the initial review in their revised manuscript and rebuttal letter. I do not have major concerns about publishing this manuscript.

Responses: Thanks very much to the reviewer for the satisfaction about concerns from the initial review and kind reminds from second review. We added the information of the equations and fitting methods related to binding events in the revised supplementary file. According to the review's suggestions, we replaced the linear fits of fluorescence/UV-Vis spectra (including Fig. 5c and 5d; Supplementary Figs. 5 and 8) with non-linear regression methods for stating more accurately. In addition, we were willing to adopt the suggestions of reviewer and acknowledge the possibility of other binding stoichiometries for the binding of NADH beyond 1:1 in the revised manuscript. We thus added several words to describe this possibility in the revised manuscript. As the systems described in Supplementary Figs. 9 and 10 (Supplementary Fig. 8 and 9 in original supplementary file) were both normal homogenous systems rather than supramolecular systems, which were applicable to intermolecular Stern-Volmer fitting method.

1. For normal homogenous system, the quenching behavior was evaluated by the Stern-Volmer equation as in Supplementary Equation (1):

$$\frac{F_0}{F} = 1 + k_q \tau_0 [Q] = 1 + K_{SV} [Q]$$

F_0 and F are the emission intensity in the absence and presence of quencher, respectively, k_q is the quenching rate constant, τ_0 is the excited-state lifetime in the absence of quencher, and $[Q]$

is the concentration of quencher. Plotting the ratio F_0/F against the quencher concentration thus gives a straight line having a y -intercept equal to 1 and a slope, termed the Stern-Volmer constant (K_{SV}), equal to $k_q\tau_0$.

2. The host-guest binding behavior showed in UV-Vis spectra was evaluated by the 1:1 binding model, as in Supplementary Equation (2):

$$A - A_0 = \frac{1}{2} (\varepsilon_{HG} - \varepsilon_H) \left[\left([G_0] + [H_0] + \frac{1}{K_a} \right) - \sqrt{\left([G_0] + [H_0] + \frac{1}{K_a} \right)^2 - 4[H_0][G_0]} \right]$$

A_0 and A are the absorbance in the absence and presence of guest, respectively, ε_{HG} is the molar absorptivity of the host-guest species, ε_H is the molar absorptivity of the free host H, $[G_0]$ is the total concentration of the guest, $[H_0]$ is the initial concentration of the host, K_a is the association constant.

3. The host-guest binding behavior showed in fluorescence spectra was evaluated by non-linear Hill plot, as in Supplementary Equation (3):

$$F - F_0 = \frac{1}{2} (F_L - F_0) \left[\left(\frac{[G_0]}{[H_0]} + 1 + \frac{1}{K_a[H_0]} \right) - \sqrt{\left(\frac{[G_0]}{[H_0]} + 1 + \frac{1}{K_a[H_0]} \right)^2 - \frac{4[G_0]}{[H_0]}} \right]$$

F_0 and F are the emission intensity in the absence and presence of guest, respectively, F_L is the emission intensity of saturated value in presence of excess guest. $[G_0]$ is the total concentration of the guest, $[H_0]$ is the initial concentration of the host, K_a is the association constant.

Following is the formula derivation:

Generally, for the formation of 1:1 host-guest species H·G formed by host (H) and guest (G), if we assume that $[H_0]$ is the initial concentration of the host, the concentration $x[H_0]$ of H·G produces when adding the molar concentration $[G_0]$ of guest. K_a can be calculated by the Supplementary Equation (4):

$$K_a = \frac{x}{(1-x)([G_0] - x[H_0])}$$

The quadratic equation can be rearranged to the Supplementary Equation (5):

$$x^2 - \left(\frac{[G_0]}{[H_0]} + 1 + \frac{1}{K_a[H_0]} \right) x + \frac{[G_0]}{[H_0]} = 0$$

The corresponding solution is shown in Supplementary Equation (6):

$$x = \frac{1}{2} \left[\left(\frac{[G_0]}{[H_0]} + 1 + \frac{1}{K_a[H_0]} \right) - \sqrt{\left(\frac{[G_0]}{[H_0]} + 1 + \frac{1}{K_a[H_0]} \right)^2 - \frac{4[G_0]}{[H_0]}} \right]$$

The measurements are performed under the conditions where the emission intensity of the free host in such a concentration is F_0 ; after addition of a given amount $[G_0]$, the fluorescent intensity can be described by the Supplementary Equation (7):

$$F = F_0(1 - x) + F_L x$$

F_L is the emission intensity of saturated value in presence of excess guest. The equation can be rearranged to the Supplementary Equation (8):

$$x = \frac{F - F_0}{F_L - F_0}$$

Combining Supplementary Equation (6) and (8) can afford the Supplementary Equation (3).

For Reviewer #2

Reviewer's Comments: The authors have done a careful and thorough job of revising the paper in light of the comments from the first three reviewers. These revisions include

- more detailed electrochemical studies
- discussion of the possibility of the cage/guest system binding on the protein external surface, including additional computational docking studies and UV/Vis spectroscopic analysis
- explicit analysis of aldehyde formation by GC
- indication of errors in experimental data
- calculation and inclusion of turnover numbers for catalysis
- confirmation of the supramolecular catalytic nature of the process using an inhibitor which blocks the cage cavity and removes most catalytic activity
- numerous small corrections to text, figures and references to clarify minor issues or lack of clarity in some cases.

The resulting extensive revision is significantly improved and I am happy to support publication. The conclusions are well evidenced and the study will attract significant interest.

Response: Many thanks to the referee for the positive and kind comments. We will try our best to improve the quality of our publishes.

**For Reviewer #3**

Comments: The authors have revised their manuscript to address a number of points raised. While the manuscript is improved, are many outstanding questions about the characterization of the system and reactivity that cause the manuscript still be speculative. Furthermore, the enzyme portion of the system is not well described and not characterized.

Reviewer's Comments (1): To better understand the Co_3TPS_2 unit's interaction with ADH, the authors have performed a docking study and how describe the Co_3TPS_2 cage as “perched on enzymatic pocket.” From a biochemical standpoint, this description of perched on a pocket does not make sense. Enzymatic pockets are buried – it sounds like, based on the docking study, part of the complex is buried and part contacting the surface near the pocket (?) It is recommended that the authors consult with biochemists or structural biologists regarding the characterization and description of this complex to use the best terms to describe this proposed interaction. The authors provide Fig. 1 with a structure of a host-guest semibiological system with the metal-organic cage placed in the ADH active site. However, there is no information in the figure caption on how this structure shown is made. Presumably, this is from the docking study, but this needs to be specified. Looking at the docking study results in the SI, they seem to show the cage associated with the enzyme surface from what I can see but it is hard to discern from these results where it is located relative to the active site as no landmarks are given. Critically, I can find no information on the origin of this ADH structure (what is the PDB code?) or assumptions made in the docking study. Regarding the experimental studies of ADH, I can't find information about how the ADH samples were obtained and characterized in this study, or, critically, even what their source is (human? yeast? what isozyme?)

Response: Many thanks to the referee for the further reminds and suggestions. After carefully consulting and referring, we considered the description that the Co_3TPS_2 cage “binding to ADH enzymatic pocket” was appropriate to illustrate the binding relationship between Co_3TPS_2 and the natural enzyme ADH. We are sorry for this careless omission about the missing of information in the Fig. 1 caption. The structure of host-guest semibiological system with the metal-organic cage placed in the pocket of ADH is from the docking study. We have added some words to describe this structure in the Fig. 1 caption. In order to clearly clarify the docking study results, we added the landmarks to show the active site of ADH in Supplementary Fig. 36. Additionally, we provided a relevant information with the docking study, in which the blind docking strategy ensured sufficient spaced to cover the entire surface of the enzyme. The PDB

code of ADH structure was 5ENV, which was described in the manuscript. We also supplied the source information of biomaterial ADH samples from *Saccharomyces cerevisiae* in the revised supplementary file.

Reviewer's Comments (2): To the best of my knowledge, ADH is a dimer (sometimes a tetramer) and the dimer is thought to be the functional unit. However, the relationship of the Co_3TPS_2 to the full ADH structure as a dimer with its two Zn sites is not described. Furthermore, there is no characterization of this structure that reveals the state of the ADH and the ADH- Co_3TPS_2 complex. A number of straightforward experiments can and should be done -- UV-vis and CD of the enzyme along compared to the enzyme with Co_3TPS_2 , hydrodynamic radius measurement (i.e. gel filtration) of enzyme and enzyme + Co_3TPS_2 , and dynamic light scattering, at least. Otherwise, the structure presented through the docking study is a proposed but not verified structure. I regret that I did not raise these issues previously, but I found the original manuscript too confusing and had difficulty figuring out what the system was that was being studied in order to make specific suggestions around characterization.

Response: Thanks very much for the thoughtful suggestions. As mentioned by reviewer, the dimer is thought to be the functional unit, which crystallized in the asymmetric form containing a closed conformation subunit and an open conformation subunit. The ADH in the docking study is a dimer pattern (PDB code: 5ENV), and the random binding model of the Co_3TPS_2 cage to ADH manifested that the cage binding to the ADH catalytic pocket was the major binding conformation with a higher binding energy (Supplementary Fig. 37). According to the reviewer's suggestions, we performed UV-Vis and CD spectra of the enzyme along compared to the enzyme with Co_3TPS_2 , and hydrodynamic radius measurement (i.e. dynamic light scattering, gel filtration chromatography) of enzyme and enzyme with Co_3TPS_2 . Both UV-Vis and CD spectra showed the characteristic peaks attributable to ADH were basically maintained after adding the cage Co_3TPS_2 (Supplementary Fig. 6a and b). Gel filtration chromatography showed that the sample of ADH with Co_3TPS_2 emerged a new peak with a shorter retention time than that of ADH, implying that Co_3TPS_2 bound to ADH giving a larger hydrodynamic radius (Supplementary Fig. 12a). The dynamic light scattering measurement of ADH showed a sharp size-distribution peak and presented an average hydrodynamic radius of approximately 6.9 nm, while an average hydrodynamic radius of approximately 7.3 nm after adding the cage Co_3TPS_2 (Supplementary Fig. 12b). These results all supports our postulation of the binding behavior of Co_3TPS_2 cage to enzyme ADH, and we added some words in the revised manuscript, and

supplied the related description.

Reviewer's Comments (3): The authors have now presented more information on the H₂ evolution activity of the Co₃TPS₂ complex, and draw conclusions on the mechanism, calling it an ECCE pathway. There is no experimental data that can support that mechanism in this paper. Indeed, distinguishing between ECEC and ECCE is usually not possible even in dedicated, detailed electrocatalysis studies of H₂ production. Further speculation that they present regarding protonation of the dithiolene species and formation of a Co(III)-H is also not supported.

Response: Thanks very much for the careful suggestions and reminds. Indeed, it was a steep challenge to distinguish the ECEC and ECCE pathways in the catalysis of the cobalt dithiolene complexes. The cobalt dithiolene complexes usually contained both these pathways in catalyzing hydrogen production as referred in the below. We also showed a sole reversible redox peak consistent with CoBDT₂ (reference 33) in CV test of Co₃TPS₂ (scanning from -1.5 to 0 V). The addition of trifluoroacetic acid triggered the appearance of a catalytic wave at approximately -0.75 and -1.20 V (Supplementary Fig. 14) being similar to reported literatures, allowing protonation to take place at either the cobalt metal or sulfur for driving hydrogen production. Meantime, the combination of NAD⁺-mediated dehydrogenation and NADH-modified hydrogen evolution is the core goal in the present study, our experimental results also demonstrated that the NADH/NAD⁺ couple could be used to communicate between the abiotic hydrogen evolution and the biotic dehydrogenation, ignoring the difference on the ECEC and ECCE pathways. Therefore, we modified the related description in the revised manuscript for a more rigorous statement.

41. McNamara, W. R., Han, Z., Yin, C., Brennessel, W. W., Holland, P. L. & Eisenberg, R. Cobalt-dithiolene complexes for the photocatalytic and electrocatalytic reduction of protons in aqueous solutions. *Proc. Natl. Acad. Sci. USA* **109**, 15594–15599 (2012).

44. Solis, B. H. & Hammes-Schiffer, S. Computational study of anomalous reduction potentials for hydrogen evolution catalyzed by cobalt dithiolene complexes. *J. Am. Chem. Soc.* **134**, 15253–15256 (2012).

In summary, the authors present a creative approach to photocatalytic H₂ production. However, there are standard, fundamental characterization steps missing in this work. Description of the ADH samples and details of the docking study setup and results are missing. Furthermore, the conclusions continue to be much too speculative (in particular regarding mechanism). As a

result, the paper continues to include greatly overinterpreted data and speculation. If the stuck to presenting their results as they are and drawing conclusions that can be well supported by the data, this would be an interesting paper worth publishing.

Response: Many thanks to the referee for the careful suggestions and reminds again. According to your suggestions, we have added experiments to address them and complete the study, and carefully revised the manuscript and supplementary file. We sincerely hope the revised manuscript will match your expectations and publish in Nature Communications.

REVIEWER COMMENTS

Reviewer #3 (Remarks to the Author):

The manuscript has now been revised to better clarify how this system works – it is a light-driven system for H₂ production by a cobalt dithiolene catalyst driven by an organic dye, where the oxidized dye reduction is achieved by NADH, and NADH is regenerated through the action of alcohol dehydrogenase, which oxidizes methanol and reduces NAD⁺. Thus, compared to “traditional” systems where something like ascorbic acid is used as the electron donor, here they are using an alcohol as the terminal electron donor. It is a clever approach, although complex. The advantage of this approach over systems that utilize direct (to the dye) sacrificial electron donors is not evident, and clarifying that in this manuscript would help.

The authors have done made many improvements in the manuscript, improving characterization and description of the system and its reactivity. Speculative statement also have been removed. I have a remaining question regarding understanding and describing the activity of the overall system.

The TON achieved (35 with respect to Co) is quite low for H₂ evolution catalysts, but the authors describe this as “comparable efficiency to those reported similar supramolecular catalytic systems that usually require the support of plenty sacrificial electron donors). However, for more simple systems, TONs can be much higher (1000’s, 10,000’s), and thus this statement overstates the activity of this system. I understand that a comparison should be made to other supramolecular systems, but the advantage of such systems is not very clear, and comparison to moare standard systems also should be made. Furthermore, “efficiency” should be reserved for description of quantum yield, not TON, which reflects a combination of durability and catalytic rate. Finally, the authors provide a TON of 875 with respect to ADH. The mismatch with the much lower value for Co deserves comment. What is the ratio of cobalt to ADH in the system? The manuscript mentions detecting aldehyde and says that H₂ and aldehyde were generated in similar amounts, which is encouraging, but how this correlates with TON values is not clear.

For Reviewer #3

Comments: The manuscript has now been revised to better clarify how this system works – it is a light-driven system for H₂ production by a cobalt dithiolene catalyst driven by an organic dye, where the oxidized dye reduction is achieved by NADH, and NADH is regenerated through the action of alcohol dehydrogenase, which oxidizes methanol and reduces NAD⁺. Thus, compared to “traditional” systems where something like ascorbic acid is used as the electron donor, here they are using an alcohol as the terminal electron donor. It is a clever approach, although complex. The advantage of this approach over systems that utilize direct (to the dye) sacrificial electron donors is not evident, and clarifying that in this manuscript would help.

The authors have done made many improvements in the manuscript, improving characterization and description of the system and its reactivity. Speculative statement also have been removed. I have a remaining question regarding understanding and describing the activity of the overall system.

The TON achieved (35 with respect to Co) is quite low for H₂ evolution catalysts, but the authors describe this as “comparable efficiency to those reported similar supramolecular catalytic systems that usually require the support of plenty sacrificial electron donors). However, for more simple systems, TONs can be much higher (1000’s, 10,000’s), and thus this statement overstates the activity of this system. I understand that a comparison should be made to other supramolecular systems, but the advantage of such systems is not very clear, and comparison to more standard systems also should be made. Furthermore, “efficiency” should be reserved for description of quantum yield, not TON, which reflects a combination of durability and catalytic rate. Finally, the authors provide a TON of 875 with respect to ADH. The mismatch with the much lower value for Co deserves comment. What is the ratio of cobalt to ADH in the system? The manuscript mentions detecting aldehyde and says that H₂ and aldehyde were generated in similar amounts, which is encouraging, but how this correlates with TON values is not clear.

Response: Many thanks to the reviewer for the kind comments and thoughtful suggestions. In this paper, we are committed to constructing a redox-neutral host–guest photocatalytic system comparable to living organisms and exploring a compatible and cooperative catalysis between abiotic and biotic catalysts. For this purpose, we followed living organisms and employed the natural coenzyme NADH as electron donor to accomplish hydrogen production half-reaction, while regenerating NADH with dehydrogenase. For example, by introducing ADH, the recycling of NADH and the conversion of alcohol to aldehyde could be simultaneously achieved within a system. The advantage of this host–guest semibiological photosynthesis approach over

traditional systems is the circumvention of sacrificial electron donors by introducing coenzyme and enzyme system, and the benign compatibility between artificial catalysts and natural enzymes by forming regional cooperation and division via the host-guest supramolecular approach. Our host-guest photocatalytic system simultaneously produced both hydrogen and aldehyde in a similar amount, which is encouraging as mentioned by reviewer, successfully achieving our original endeavor to synergistically catalyze with natural enzymes in an efficient and compatible way by means of supramolecular catalysts. We have added some words to clarify the advantage of this host-guest semibiological photosynthesis system.

Actually, the TONs for H₂ evolution catalysts are affected by different types and different concentrations of photosensitizers and electron donors. Researchers usually optimize the reaction conditions to obtain an ideal TON value with respect to the least catalytic component. For traditional systems (for example reference 33/41 in manuscript), by excessively adding the well-matched photosensitizers (>100 times to catalysts) and electron donors (>10000 times to catalysts) with catalysts, the TON values with respect to catalysts can be much higher (>2000), while the TON values with respect to photosensitizers (>20) are much lower than that with respect to catalysts. The huge concentration difference between photosensitizers and catalysts resulted in the mismatch of TONs with respect to photosensitizers and catalysts, and the catalytic TON values for catalysts decreased with the drop of the concentration of photosensitizers and electron donors (see reference 32/33). Meanwhile, the catalyst CoBDT₂ used in reference 33 showed an inferior catalytic activity compared to supramolecular catalyst Co₃TPS₂ when applying to the ternary photocatalytic hydrogen evolution system with NADH as electron donor (Supplementary Table S3, entry 1 and entry 8). In addition, CoBDT₂ was incompatible with natural enzymes, while the hybrid system consisting of metal-organic cages Co₃TPS₂ and natural enzymes could perfectly synergistically catalyze hydrogen evolution and alcohol dehydrogenation pursuing a redox-neutral system (Fig 3b). Therefore, we think it is inappropriate to compare the TON value between our tandem system and the traditional systems in this study.

Consistent with traditional homogeneous catalytic systems, the TON values were defined as the ratio of the molar amount of hydrogen generated to that of cobalt on Co₃TPS₂ or ADH in the original manuscript. Through optimizing reaction conditions of host-guest photocatalytic system, the ratio of ADH to cobalt in the typical system is 4% for ensuring the catalytic performance of enzymes (Fig. 4). In this condition, cobalt content is excessive to ADH, therefore, resulting in the inconsistent TONs with respect to Co and ADH. In order to clearly evaluate our system, we only showed the TONs of hydrogen and aldehyde with respect to ADH to reflect the catalytic activity

大连理工大学

精细化工国家重点实验室

State Key Laboratory of Fine Chemicals

of the overall system in the revised manuscript. Moreover, according to the reviewer's suggestions, we corrected the inappropriate express "efficiency" to "catalytic activity" in the revised manuscript to describe the catalysis more accurately, and the relationship between the aldehyde amounts and TON values was also described at the corresponding positions. All changes in the manuscript are marked in light yellow in the revised manuscript.